# Framework engineering to produce dominant T cell receptors with enhanced antigen-specific function

Sharyn Thomas[1], Fiyaz Mohammed[2], Rogier M. Reijmers [3], Annemarie Woolston [1], Theresa Stauss[1], Alan Kennedy [1], David Stirling [1], Angelika Holler[1], Louisa Green [1], David Jones [4], Katherine K. Matthews[5], David A. Price [5], Benjamin M. Chain [1], Mirjam H.M. Heemskerk [3], Emma C. Morris [1], Benjamin E. Willcox [2] & Hans J. Stauss [1]\*

TCR-gene-transfer is an efficient strategy to produce therapeutic T cells of defined antigen specificity. However, there are substantial variations in the cell surface expression levels of human TCRs, which can impair the function of engineered T cells. Here we demonstrate that substitutions of 3 amino acid residues in the framework of the TCR variable domains consistently increase the expression of human TCRs on the surface of engineered T cells. The modified TCRs mediate enhanced T cell proliferation, cytokine production and cytotoxicity, while reducing the peptide concentration required for triggering effector function up to 3000-fold. Adoptive transfer experiments in mice show that modified TCRs control tumor growth more efficiently than wild-type TCRs. Our data indicate that simple variable domain modifications at a distance from the antigen-binding loops lead to increased TCR expression and improved effector function. This finding provides a generic platform to optimize the efficacy of TCR gene therapy in humans.

[1] Institute of Immunity and Transplantation, Division of Infection and Immunity, University College London, Royal Free Hospital, London NW3 2PF, UK. [2] Cancer Immunology and Immunotherapy Centre, Institute for Immunology and Immunotherapy, University of Birmingham, Edgbaston, Birmingham B15 2TT, UK. [3] Department of Hematology, Leiden University Medical Center, 2300 RC Leiden, The Netherlands. [4] Department of Computer Science, University College London, London WC1E 6BT, UK. [5] Division of Infection and Immunity, Cardiff University School of Medicine, Cardiff CF10 3AT, UK. \*email: h.stauss@ucl.ac.uk

T cell receptor (TCR) gene transfer is an effective strategy to produce therapeutic T cells with clinical benefit in the treatment of cancer[1–3]. The ability of engineered T cells to be stimulated by low concentration of peptide antigen is a key parameter for the efficacy of TCR gene therapy. Hence, mutagenesis of the antigen-binding regions CDR1, 2, and 3 has been employed to select TCRs with enhanced affinity for cancer-associated peptide antigens[4–10]. However, in vitro studies have revealed that TCRs with affinities below 450 nM retained peptide specificity, while higher TCR affinities were associated with loss of specificity and increased reactivity against cells that did not express the cognate target antigen[6,7]. In addition, high-affinity interactions can disrupt serial TCR triggering which facilitates T cell stimulation at low antigen concentrations[11]. Consequently, TCRs with super-physiologically high affinities can fail to trigger T cell responses at low peptide concentrations[12].

Improving the surface density of TCR without changing TCR affinity is an alternative strategy to enhance the avidity and function of therapeutic T cells. To date, modifications in the TCR constant regions, including the introduction of additional cysteine di-sulfide bonds and sequences of murine origin, have been used to improve human TCR α/β chain pairing and expression[13–17]. However, human TCRs with identical constant regions show large differences in surface expression, indicating a major role of the variable (V) α and Vβ domains in TCR assembly[18,19].

In this study, we explore whether residues in the framework of the Vα and Vβ domains determine the efficacy of intracellular TCR assembly and the level of surface expression. In addition, we test the hypothesis that amino acid replacements in the Vα and Vβ domains outside the antigen-binding CDR loops can be exploited to enhance antigen-specific T cell effector function without disturbing the fine specificity of TCRs.

## Results

**Dominant and subdominant TCR repertoire.** In order to identify dominant and subdominant TCRs in the natural human repertoire we transduced human peripheral blood T cells with synthetic TCRs that were engineered to achieve dominant expression[20]. These synthetic TCRs were codon-optimized and equipped with murine constant regions containing an additional disulfide bond to enhance α/β chain pairing (Fig. 1a). Surface expression of the introduced synthetic TCRs was assessed with anti-murine constant region antibodies, and the expression of endogenous 'natural' TCRs was assessed with anti-human constant region antibodies. Using three different synthetic TCRs we found a population of T cells that co-expressed both the introduced and endogenous TCR and a population that expressed only the introduced TCR (Fig. 1b; Supplementary Fig. 1a). This profile allowed us to define 'dominant' endogenous TCRs that were co-expressed with the synthetic TCR, and 'weak' endogenous TCR that were unable to compete for cell surface expression with the synthetic TCR. Untransduced T cells expressed only the endogenous TCR (Fig. 1b and Supplementary Fig. 1a, bottom right quadrant).

We used flow cytometry to purify T cells with dominant and weak endogenous TCRs, followed by TCR repertoire analysis (Fig. 1c). From three different donors we used Sanger sequencing to generate a sequence library containing 884 distinct TCR clonotypes, half with a dominant and half with a weak expression phenotype. Analysis of variable domain usage showed that TRAV38-1, TRAV38-2, TRBV5-1, and TRBV7-8 were significantly enriched in the dominant TCR library, whereas TRAV13-2, TRBV9, TRBV7-9, and TRBV2 were over-represented in the weak TCR library (Fig. 1d). Significantly increased frequencies of particular amino acids were observed at certain positions in the

dominant TCRs (Supplementary Table 1). We also employed next generation sequencing to generate much larger TCR libraries from two additional donors[21]. These libraries contained more than 130,000 distinct clonotypes, and the statistical analysis revealed enrichment of additional amino acid residues that were not detected in the small TCR library. The Vα and Vβ domains have 77 residues located in the framework regions outside the antigen-binding CDR loops. The Vα analysis showed over-representation of certain amino acids at 63 of the 77 framework positions, and the Vβ analysis revealed over-representation at 68 of 77 positions. In order to identify candidate positions for experimental testing we used structural modeling and selected 14 residues for detailed functional analysis (Fig. 1e). The selected residues were located in the framework regions and fell into one of four categories: (1) solvent exposed; (2) hydrophobic core; (3) Vα–Vβ interface; and (4) Vα–Cα or Vβ–Cβ interface (Fig. 1e).

**Cell surface expression levels TCRα/β chains.** We designed a retroviral vector containing V5 and myc tags to quantify surface expression levels of the TCR α and β chains using V5/myc-specific antibodies (Fig. 2a). The vector also contained the truncated murine CD19 marker molecule to monitor transduction efficacy. First we compared the expression of a TCRα/β sequence (TRAV32-8/TRBV7-8) that was enriched in our dominant TCR library with the expression of three TCRα/β sequences that were enriched in our weak TCR library (Fig. 1d). Transduction of human Jurkat cells, which expressed endogenous CD3 and TCR, indicated that dominant TCR α/β chains were expressed in a higher percentage of cells than weak TCR α/β chains (Fig. 2b). In addition, the expression level of the weak TCRs was ~3–5-fold lower compared to the expression of the dom TCR α/β chains (Fig. 2c). This difference was observed when gating on Jurkat cells expressing high or intermediate levels of the CD19 marker used to monitor transduction efficacy. Similar results were obtained with Jurkat cells lacking endogenous TCR (Supplementary Fig. 2a), indicating that the surface expression of dominant TCR is superior to that of weak TCR in the presence, and also in the absence of competition from endogenous TCR.

Next, we tested whether the 14 candidate residues indicated in Fig. 1e affected the level of TCR expression. Replacement of all 14 residues converted a weak TCR into a 'dominant' TCR (weak → domTCR) by improving expression levels by more than 7-fold (Fig. 2d, e). In contrast, replacing these residues in the dominant TCR with the amino acids found in the weak TCR dramatically reduced expression of the converted dom → weak TCR to undetectable levels (Fig. 2d, e). A similar impact of the 14 residues on TCR expression was observed in Jurkat cells lacking endogenous TCR (Supplementary Fig. 2b).

Subsequent experiments were designed to test the impact of individual residues on TCR expression. The results demonstrated that the change of proline at position 96 of the weak α chain (P96α) to leucine (L96α), or a double amino acid change from serine/asparagine (S9β/N10β) to arginine/tyrosine (R9β/Y10β) at position 9 and 10 of the β chain resulted in nearly three-fold increase in TCR surface expression (Fig. 3a, b). We further tested biochemically similar amino acids at the same positions. Supplementary Fig. 3 shows that a hydrophobic amino acid at position 96α was sufficient to improve TCR expression on the cell surface. Similarly, biochemically equivalent amino acids at position 9β and 10β had similar effects on TCR expression. The data also revealed that position 10 of the β chain had a stronger effect on TCR expression than position 9 (Supplementary Fig. 3).

The introduction of valine at position 19α (V19α) and threonine at position 24α (T24α) also improved TCR expression,

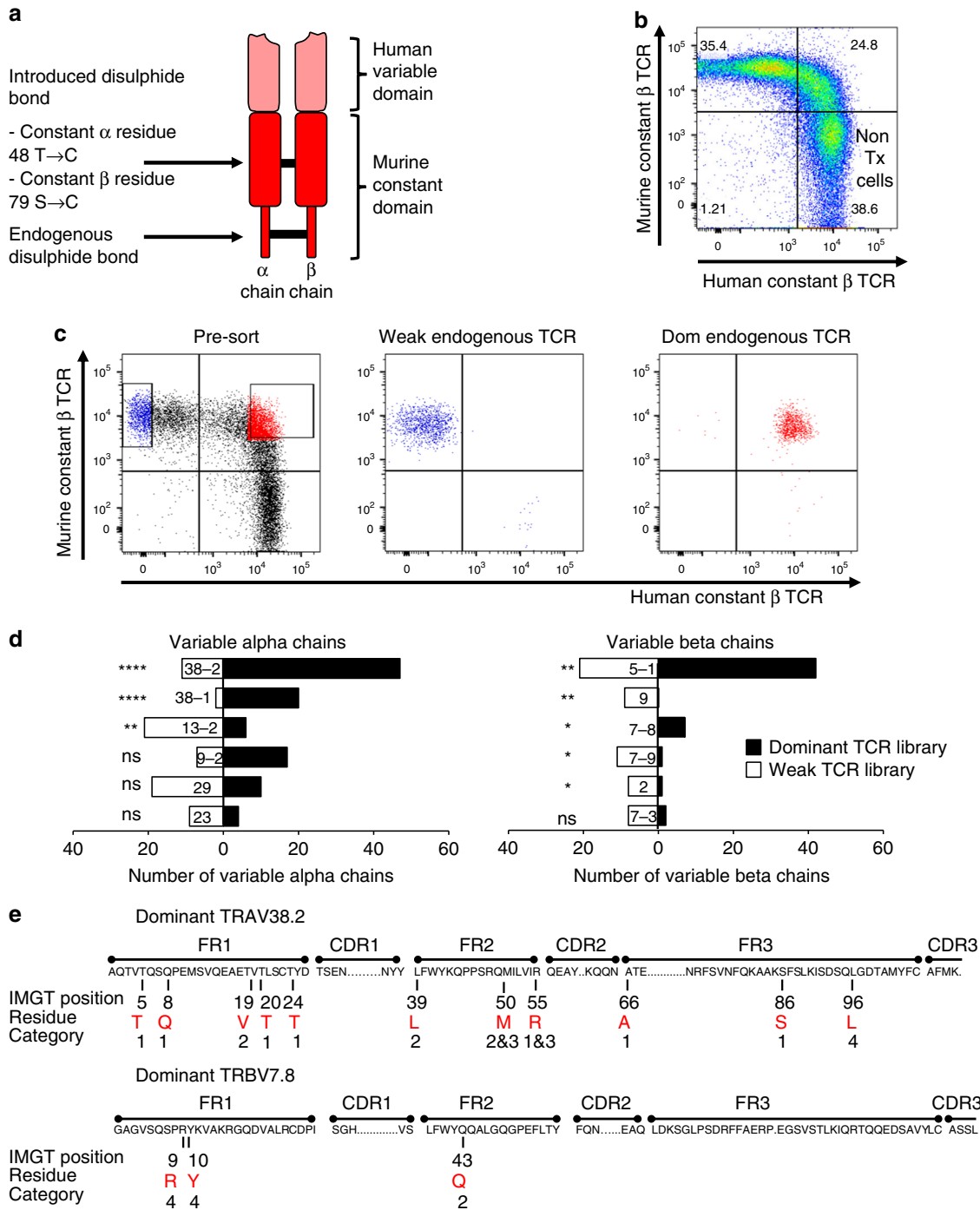

**Fig. 1** Identification of dominant and weak human TCRs. **a** Schematic representation of a synthetic dominant TCR containing codon-optimized human variable domains and codon-optimized murine constant domains with an additional disulfide bond (C48α and C79β; IMGT nomenclature). **b** Dot plot of polyclonal peripheral blood human T cells transduced with a synthetic dominant TCR specific for WT1 and double-stained with anti-human constant domain antibodies to identify the endogenous TCR and anti-murine constant domain antibodies to identify the introduced TCR. Live, single cells were first gated on CD3. Non-transduced (non-Tx) T cells are labeled. **c** Transduced T cells expressing weak endogenous TCRs and transduced T cells expressing dominant endogenous TCRs were purified by flow cytometry. An unbiased molecular approach was then used to identify all expressed TCRs. The experiment was repeated independently with $n = 3$ different donors, and the TCR-sequencing data were pooled. **d** T cells receptor variable gene segment frequencies in the library of dominant TCRs (solid black bars) and the library of weak TCRs (open bars). *$P < 0.05$; **$P < 0.01$; ****$P < 0.0001$; ns, $P > 0.05$ (unpaired $t$-test). **e** Candidate residues in the framework regions of the Vα and Vβ domains that were selected for detailed studies based on structural analyses and occurrence at high frequencies in the dominant TCR library. Candidate residues are shown in red letters. Numbers above the residues denote the IMGT positions. The category numbers indicate the position of residues in the TCR structure. 1. Solvent exposed; 2. hydrophobic core; 3. Vα–Vβ interface; 4. Vα–Cα or Vβ–Cβ interface. FR framework region, CDR complementarity-determining region

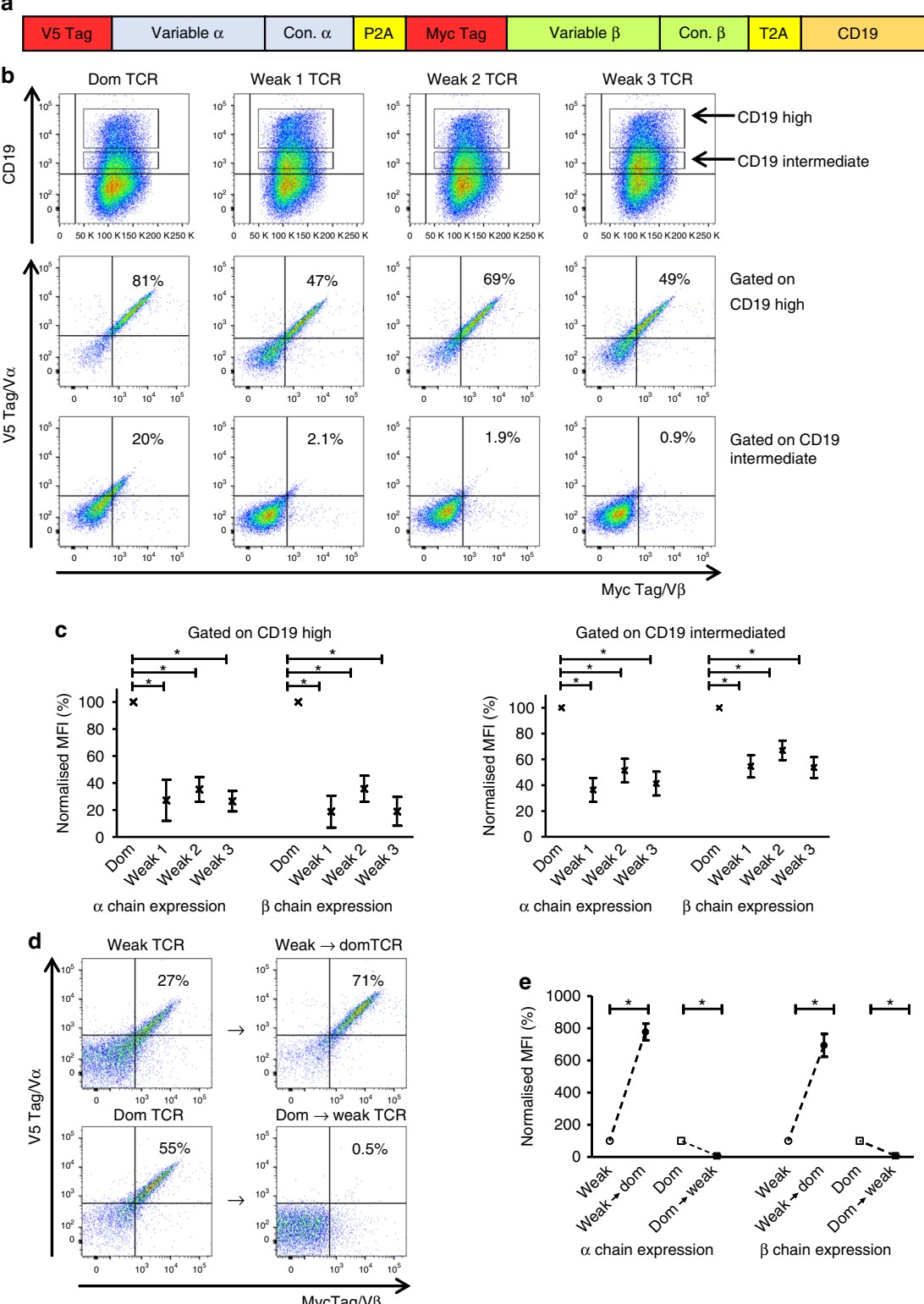

but to a lesser extent than L96α and R9β/Y10β. In contrast, introducing L39α and R55α as well as Q43β, dramatically reduced expression of the weak TCR (Fig. 3a). The detrimental effect of these three amino acids on surface expression was clearly context dependent, because the same residues were present in the well-expressed dominant TCR, or the converted weak → domTCR generated by introducing all 14 candidate residues, including L39α, R55α, and Q43β, into the weak TCR (Fig. 2d). It was

therefore possible that these three residues were suboptimal and impeded maximal expression of the dominant TCR. To test this hypothesis, we introduced the weak TCR amino acids F39α, D55α, and R43β into the dominant TCR and the converted weak → domTCR. These changes further improved surface expression (Fig. 3c, d). Accordingly, not all amino acid residues found in dominant TCR chains contribute equally to optimal expression.

**Fig. 2** Conversion of a weak TCR into a dominant TCR by replacement of 14 variable region framework residues. **a** Schematic representation of the retroviral vector used for TCR expression studies. TCR α and β chain expression was determined using antibodies specific for the V5 and myc epitopes, respectively. Transduction efficiency was determined using antibodies specific for murine CD19. **b** Representative example of $n = 4$ independent experiments showing human Jurkat cells (expressing an endogenous TCR) transduced with a dominant (Dom) TCR (TRAV38-2/TRBV7-8) or three different weak TCRs: weak 1 (TRAV13-2/TRBV7-3), weak 2 (TRAV23/TRBV7-9) or weak 3 TCR (TRAV29/TRBV2). Top panel: CD19 expression levels. Middle panel: TCR α and β chain expression levels on gated CD19[high] cells. Bottom panel: TCR α and β chain expression levels on gated CD19[intermediate] cells. **c** Pooled data (means ± SEM) showing TCR α and β chain expression levels normalized to the Dom TCR. $n = 4$ independent experiments. Top panel: gated on CD19[high] cells. Bottom panel: gated on CD19[intermediate] cells. * $P < 0.05$ (Mann–Whitney $U$ test) for all comparisons between the Dom TCR α chain and the weak TCR α chains and for all comparisons between the Dom TCR β chain and the weak TCR β chains. MFI, median fluorescence intensity. **d** Top panel: introduction of the 14 residues indicated in Fig. 1e into the weak 1 TCR (TRAV13-2/TRBV7-3) generated the weak → dom TCR with enhanced α/β expression on the cell surface. Bottom panel: replacement of the 14 residues in the Dom TCR (TRAV38-2/TRBV7-8) with the equivalent residues in the weak 1 TCR (TRAV13-2/TRBV7-3) generated the dom → weak TCR with undetectable α/β expression on the cell surface. TCR constructs were transduced into Jurkat cells expressing an endogenous TCR. Data are representative of four independent experiments. **e** Pooled data (means ± SEM) showing TCR α and β chain expression levels normalized to the corresponding unmodified TCRs. $n = 4$ independent experiments. *$P < 0.05$ (Mann–Whitney $U$ test) for all comparisons between the modified TCRs and the corresponding unmodified TCRs. MFI median fluorescence intensity. Vα variable alpha, Vβ variable beta

**RNA and intracellular protein expression levels**. To gain molecular insights into these effects, we used a prime flow RNA assay to quantify intracellular TCR α/β mRNA and surface TCR protein levels simultaneously. Intracellular levels of α/β mRNA were similar for all TCRs analyzed and did not correlate with surface levels of the respective proteins (Fig. 4a, b). For example, the converted dom → weak TCR was not detectable on the cell surface, despite high expression levels of the corresponding α/β mRNA. We then used confocal microscopy to compare the level of intracellular protein of the two TCR constructs with the highest surface expression (weak → dom TCR) and with the lowest surface expression (dom → weak TCR). Conformation-independent antibodies specific for the V5 tag were used to assess the amount of intracellular α chain protein irrespective of TCR folding and pairing. Jurkat cells transduced with either construct expressed intracellular TCRs (Fig. 4c). Single cell quantification of the confocal data indicated an association between the intensity of staining for CD19 and the intensity of staining for V5, and suggested that the amount of α chain was slightly higher for the weak → dom TCR compared with the dom → weak TCR (Fig. 4d). However, comparison of the confocal profile and the surface expression profile determined by flow cytometry (Fig. 4e) demonstrated that a lack of TCR on the cell surface was not caused by a lack of intracellular TCR. This finding suggested that inefficient protein folding and/or suboptimal assembly of the α and β chains impaired surface expression of the dom → weak TCR.

**Modeling of TCR structure**. We next used TCR structural modeling to explore in detail how residues in the framework of the variable domains might affect TCR stability. In particular, we explored the molecular mechanism by which amino acids changes in a weak TCR resulted in enhanced surface expression. The weak TCR that was most extensively tested in our study consisted of TRAV13-2 and TRBV7-3 (Fig. 1d). Since the structure of the TRAV13-2 chain is yet to be determined, we performed our modeling using TRAV13-1 (PDB code 3PL6[22]), which is closely related to TRAV13-2. The 3PL6 TCR structure consists of the TRAV13-1 chain paired with TRBV7-3, the same chain that is present in our weak TCR. The 14 variable domain residues that were analyzed in this study were mapped onto the weak TCR structure (Fig. 5a).

The change of P96α to L96α resulted in three-fold increase in TCR expression (Fig. 3a, b). This residue protrudes from the short 3₁₀ helix that precedes strand F and packs against the Cα domain. Modeling of L96α in TRAV13-1 revealed additional hydrophobic packing interactions with non-polar residues of the Cα interface (e.g. V161α and P112α) relative to the weak TCR

(Fig. 5b). Thus, the improved TCR expression achieved with L96α is possibly due to enhanced stability of the Vα–Cα interface, although 3D structural data will be required to confirm this.

Substitutions at position 9β and 10β also enhanced TCR expression (Fig. 3a, b). In TRBV7-3 the S9β and N10β residues mediate minimal interactions at the Vβ–Cβ interface (Fig. 5c). Modeling the replacement of S9β with R9β shows that it could form a potential salt bridge with neighboring E159β (Cβ domain) (Fig. 5c). Also, Y10β is predicted to form stacking interactions with residues that protrude from the Cβ domain (Y218β and H157β), which is likely to increase the stability of the Vβ–Cβ interface relative to the native TRBV7-3 (Fig. 5c). Thus, the improved TCR expression achieved by the introduction of R9β and Y10β is likely caused by the enhanced stability of the Vβ–Cβ interactions.

The positive effect of V19α on TCR expression can be explained by its protrusion from strand B into the hydrophobic core. Replacing S19α with V19α shows that the side chain of valine could stabilize the hydrophobic core by mediating multiple non-polar interactions with L11α (strand A), V13α (strand A), I21α (strand B), and I91α (strand E) (Fig. 5d). Therefore, the V19α substitution enhances the stabilization of the hydrophobic core of the Vα domain. Finally, A24α is a solvent exposed residue that protrudes from strand B. Modeling the replacement of A24α with T24α suggests that this residue is likely to mediate a hydrogen bonding interaction with the imidazole ring nitrogen of H86α (strand E) (Fig. 5e). Thus the generation of a new hydrogen bond can provide a molecular explanation of the improved TCR expression mediated by T24α (Fig. 3a, b).

The modeling data above indicated that efficient interactions between the variable and constant domains in the α and β chain were particularly important for high-level TCR expression. We therefore used eight different TCRs to test whether improving the variable/constant interaction consistently enhanced expression of TCRs irrespective of V-region usage and specificity. The data showed that the combination of L96α, R9β, and Y10β did indeed enhance the expression of all TCRs tested (Fig. 6a, b). Consistent with previous reports the wild-type CMV2-TCR was an extreme case of a poorly expressed TCR that was undetectable on the cell surface[18], which was similar to the profile observed when the dominant TCR was converted into the poorly expressed dom → weak TCR (Fig. 2d). The triple LRY modification of the CMV2-TCR resulted in surface expression, although at lower levels than the other TCRs. Comparison of all wild-type and LRY-modified TCRs indicated that the replacement of three amino acids consistently increased surface expression in Jurkat cells by ~2–6 fold (Fig.6a, b), irrespective of the presence or absence of endogenous TCR (Supplementary Fig. 4a, b).

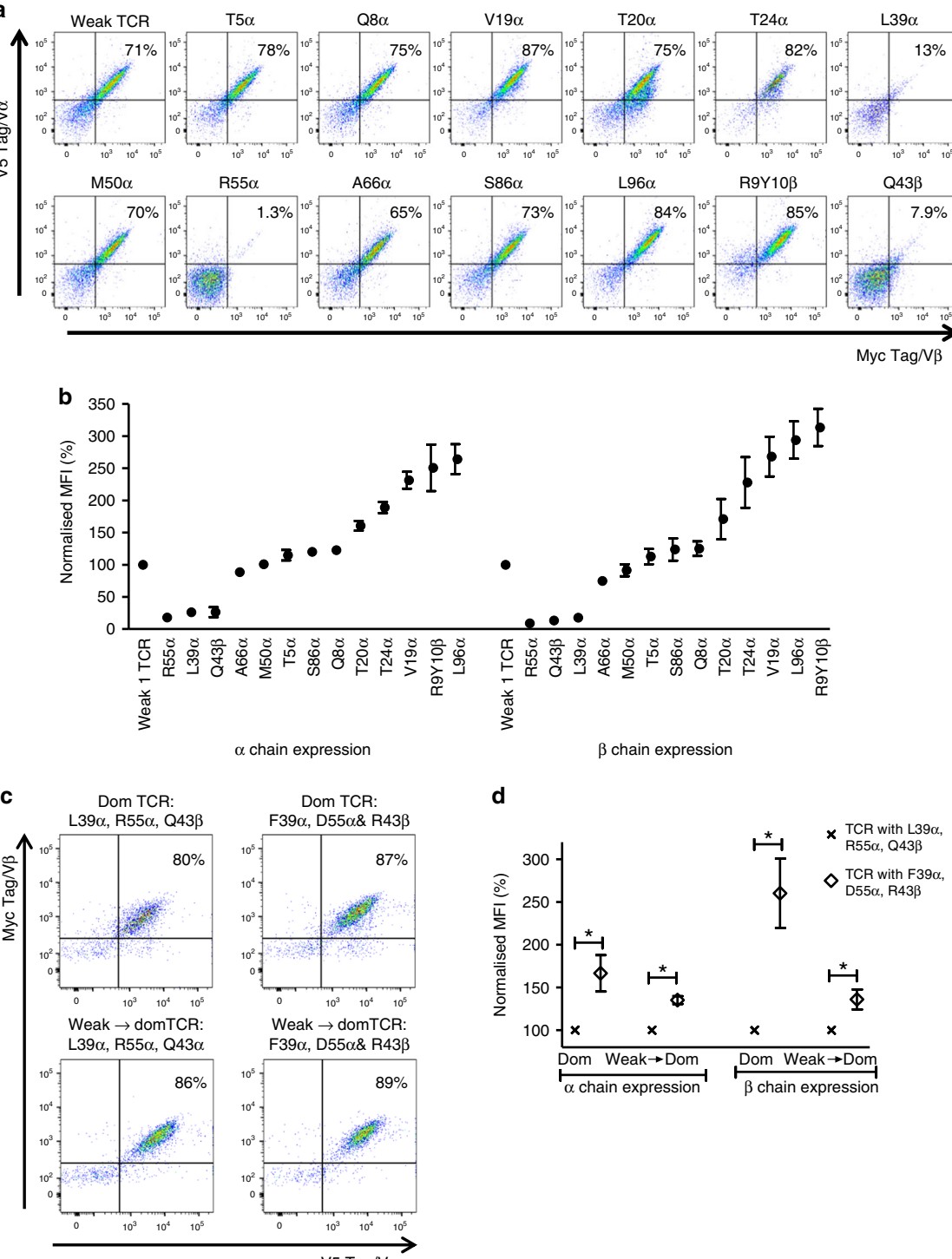

**Function of modified TCRs**. Next, we analyzed in detail the expression and function of four antigen-specific TCRs in primary human T cells. The presence of the V5 and myc tags provided a powerful tool to distinguish the introduced TCR chains from the endogenous human TCR. More importantly, the V5/myc-staining profile enabled us to identify double-positive T cells expressing both introduced TCR chains, and also single-positive T cells expressing only one of the introduced chains mis-paired with an endogenous TCR chain. Interestingly, there is a clear association between the level of TCR expression in Jurkat cells (Fig. 6a) and the frequency of primary human T cells expressing both TCR

chains (Fig. 6c). For example, the wild-type HA1.m7 TCR is poorly expressed in Jurkat cells and displays high levels of mis-pairing in primary T cells, as 41% of cells express only the introduced α chain, 12% only the β chain, and 24% both α/β chains. The analysis of all four TCRs showed that the LRY modification increased the frequency of primary T cells expressing both TCR α/β chains from a range of 20–50% to 50–80% (Fig. 6d). This effect was confirmed using tetrameric antigen complexes (Fig. 6e), which also stained primary T cells transduced with LRY-modified TCRs more intensely than primary T cells transduced with wild-type TCRs (Fig. 6f).

**Fig. 3** Single amino acid replacements in the framework regions of the Vα and Vβ domains can enhance TCR expression. Site-directed mutagenesis was used to introduce single amino acids present in the framework regions of the dominant TCR (TRAV38-2/TRBV7-8) into the framework regions of the weak 1 TCR (TRAV13-2/TRBV7-3). **a** Representative example of four independent experiments showing Jurkat cells transduced with constructs encoding the unmodified weak 1 TCR or mutated variants of the weak 1 TCR with changes in the indicated framework residues of the Vα and Vβ domains. The dot plots show TCR α/β expression levels on gated Jurkat cells expressing equivalent levels of CD19. **b** Pooled data (means ± SEM) showing how individual residues affected TCR α and β chain expression levels in Jurkat cells. Normalized to the weak 1 TCR. $n = 4$ independent experiments. $P$ values were less than 0.05 for most comparisons between the mutated variants and the weak 1 TCR (Mann–Whitney $U$ test). $P$ values were more than 0.05 (ns) for M50α and T5α with respect to α chain expression and for M50α, T5α, S86α and T20α with respect to β chain expression (Mann–Whitey $U$ test). MFI median fluorescence intensity. **c** The L39α, R55α and Q43β residues present in the dominant (Dom) TCR (TRAV38-2/TRBV7-8) were replaced with the F39α, D55α and R43β residues present in the weak 1 TCR (TRAV13-2/TRBV7-3). Similarly, the F39α, D55α and R43β residues were introduced into the weak → dom TCR (Fig. 2d) to replace L39α, R55α and Q43β. The dot plots show TCR α/β expression levels on gated Jurkat cells expressing equivalent levels of CD19. Data are representative of four independent experiments. **d** Pooled data (means ± SEM) showing how residues F39α, D55α and R43β affected TCR α and β chain expression levels in Jurkat cells. Normalized to the unmodified TCRs. $n = 4$ independent experiments. *$P < 0.05$ for all comparisons between the modified TCRs and the corresponding unmodified TCRs (Mann–Whitney $U$ test). MFI median fluorescence intensity. Vα variable alpha, Vβ variable beta

In order to determine if the LRY modification enhanced antigen-specific function, we stimulated TCR-transduced primary T cells with cognate peptides and measured the intracellular production of IFNγ and IL-2, after gating for equivalent expression levels of CD19. The LRY modification resulted in at least two-fold increase in the percentage of T cells producing IL-2 and/or IFNγ in response to stimulation with saturating concentrations (10 μM) of peptide antigen (Fig. 7a, b). There was a correlation between LRY-mediated increase in TCR expression levels on the surface (Fig. 6b) and the relative increase in antigen-specific cytokine production (Fig. 7b). Using multiplex analysis we also tested the Th1, Th2, and Th17 cytokines secreted by T cells expressing wild-type and LRY-modified TCRs. We found that the LRY modification increased the production of all 13 cytokines tested. This included the Th1 cytokines IL-2, IFNγ, and the Th2 cytokines IL-4, IL-5, IL-6, IL9, IL-10, and the Th17 cytokines IL-17A, IL-17F (Supplementary Fig. 5a). Together, this suggested that the LRY modification did not preferentially enhance Th1, Th2, or Th17 cytokine production. In addition, the LRY modification substantially enhanced antigen-specific proliferation, measured using a cytoplasmic dye (Fig. 7c), cytotoxicity, determined by a flowcytometry assay (Fig. 7d), and activation, measured using the surrogate marker CD69 (Fig. 7e).

When T cells were stimulated with titrated peptide concentrations (10 μM–1 nM) the LRY modification substantially improved the dose response profile (Fig. 7f and Supplementary Fig. 5b). For example, stimulation of the LRY-modified CMV1 TCR with 3 nM cognate peptide resulted in more IFNγ and IL-2 production than stimulation of the wild-type TCR with a 3000-fold higher peptide concentration of 10 μM. A similar improvement was observed for the HA1.m2 TCR, while the HA2.19 TCR benefitted less from the LRY-modification, most likely due to relatively high expression levels of the wild-type TCR (Fig. 7f). Together, these data indicate that the simple LRY-modification improved T cell avidity substantially, which enabled robust antigen-specific immune responses at low concentrations of peptide antigen.

To determine if the LRY-modification altered the TCR fine specificity, we generated variants of the nine amino acid long peptide epitope that is recognized by the CMV1-TCR. Each native residue was replaced with alanine, except for the variant at position 7, where the native alanine was replaced with serine. These variants were used to stimulate transduced T cells expressing the wild-type CMV1-TCR or the LRY-modified TCR. The peptide-specific IFNγ production against each variant peptide was assessed relative to the maximal response seen with the unmodified cognate peptide (Fig. 7g). Identical analyses were performed for the wild-type and LRY-modified versions of the HA1.m2 and the HA2.19 TCRs (Supplementary Fig. 6). No significant differences in cross-reactivity were observed between any of these paired wild-type and LRY-modified TCRs.

**Tumor protection by modified TCRs**. In a final series of experiments, we used a preclinical xenograft model to study the effect of LRY modification on tumor control in vivo. Mice were inoculated with U266 multiple myeloma cells, expressing HLA-A*0201 and the HA1 minor histocompatibility antigen that is recognized by the HA1.m7 TCR. After injection, U266 cells were allowed to form established tumors in the bone marrow for 10 days, and mice were then treated with engineered CD8+ T cells transduced to express the wild-type HA1.m7 TCR, the LRY-modified HA1.m7 TCR or the control CMV1 TCR. On day 0 of treatment all mice displayed similar tumor burden as determined by bioluminescent imaging (Fig. 8a). All groups of mice received the same number of transduced CD8+ T cells, determined according to surface expression of CD19. Bioluminescent images taken at day 14 showed that mice treated with the LRY-modified HA1.m7 TCR had the lowest tumor burden (Fig. 8a). The imaging data collected over the 28-day period after T cell transfer showed that the LRY-modified HA1.m7 TCR was significantly more potent at inhibiting tumor growth in vivo than the wild-type TCR (Fig. 8b).

## Discussion

We have dissected the molecular basis by which the variable domain framework regions determine the efficacy of TCR assembly and surface expression. In each TCR chain, the interactions between the variable and constant domains were particularly important for efficient surface expression. Replacement of one suboptimal residue at the Vα–Cα interface and two suboptimal residues at the Vβ–Cβ interface consistently increased TCR expression by approximately three-fold on the surface of transduced cells. In keeping with previous reports, we found that strong versus poor surface expression is an intrinsic TCR feature seen in the presence, but also in the absence of competition from endogenous TCRs[18]. This strongly suggests that the driving force of TCR evolution was the generation of diversity at the expense of optimal assembly and surface expression. We speculate that thymic repertoire selection may function to adjust differences in TCR expression levels. Thymocytes with poorly expressed TCRs may not reach the avidity threshold for positive selection and preferentially die by neglect, while thymocytes with strongly expressed TCRs may exceed the avidity threshold for negative selection resulting in their preferential deletion from the selected repertoire.

We used a relatively small library of 884 TCR clonotypes with dominant and weak expression phenotypes. The modeling of the amino acid positions in the 3D TCR structure was employed to select a set of 14 candidate residues for detailed functional studies. Although we identified particular residues with dramatic effects on TCR expression, it was possible that we overlooked certain

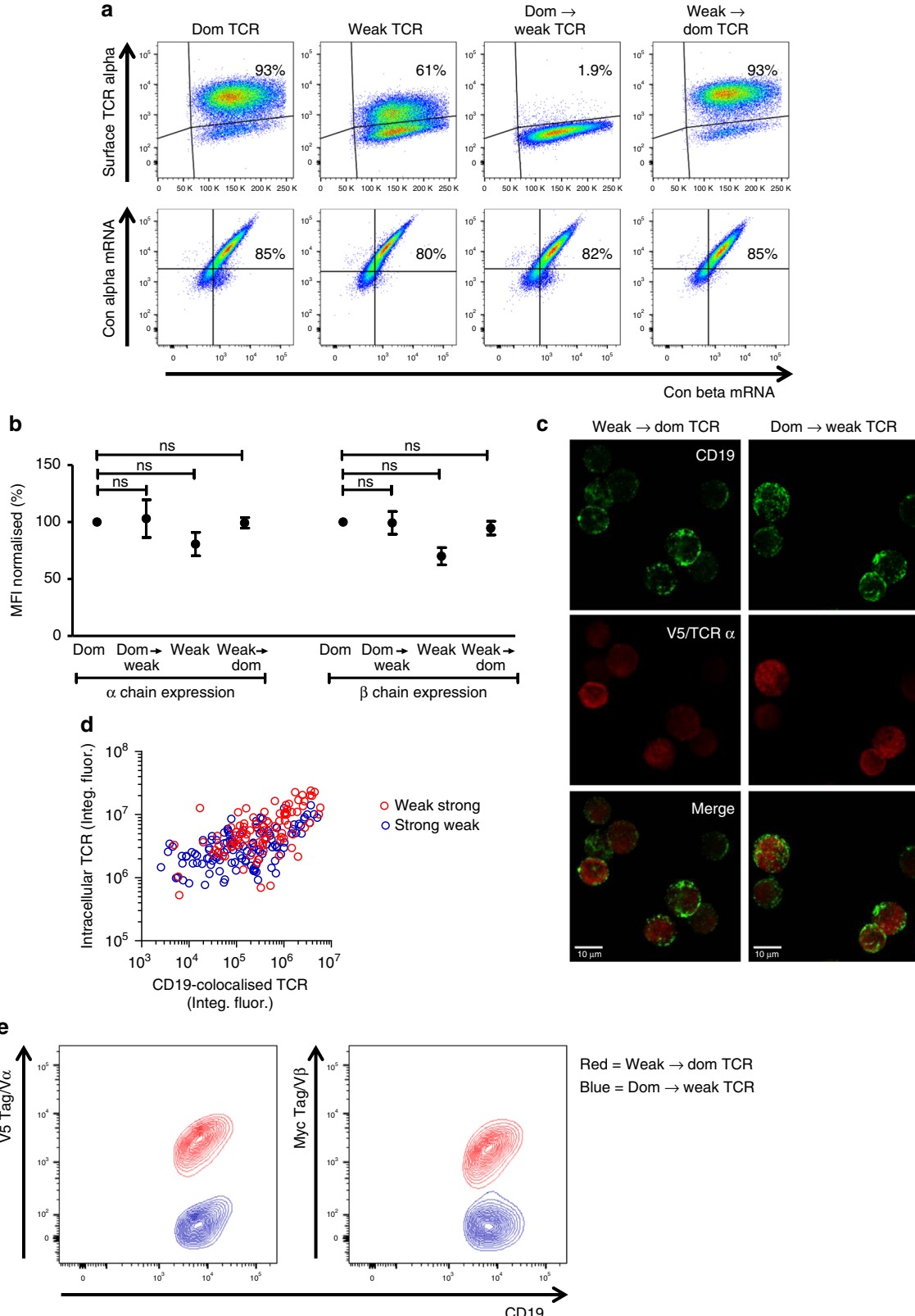

residues with more subtle effects on TCR expression. In order to explore this, we have used next generation sequencing to generate large libraries containing more than 130,000 α/β TCR clonotypes with dominant and weak expression phenotypes. Bioinformatic analyses of these large TCR libraries confirmed the importance of hydrophobic residues at position 96 of the α-chain, and of arginine and tyrosine at positions 9 and 10 of the β-chain (Supplementary Table 1). Moreover, a number of additional amino acid residues were significantly more frequent in the dominant TCR library compared with the weak TCR library. However, it seems likely that the role of these additional residues is relatively subtle, because the LRY modification alone was

**Fig. 4** Dominant and weak TCRs have similar intracellular mRNA and protein expression levels. Jurkat cells were transduced with the four TCR constructs used in Fig. 2d. **a** Top panel: cells were stained for V5 to determine TCR α chain expression on the cell surface. Bottom panel: a prime flow assay was used to quantify intracellular TCR α/β mRNA. Cells were gated for equivalent expression of CD19. Data are representative of three independent experiments. **b** Pooled data (means ± SEM) showing α/β mRNA levels normalized to the dominant (Dom) TCR. $n = 3$ independent experiments. ns (non-significant), $P > 0.05$ (Mann–Whitney $U$ test) for all comparisons between the dominant TCR and the weak TCRs. MFI median fluorescence intensity. **c** Jurkat cells were transduced with the weak → dom TCR or the dom → weak TCR and sorted by flow cytometry to purify CD19$^{high}$ cells. These cells were permeabilized and stained with anti-CD19 (green) and anti-V5/TCR α (red) for analysis via confocal microscopy. Bottom panel: overlay of CD19 and V5/TCR α expression. Scale bars are 10 μM. **d** Single cell analysis of the confocal data showing the quantification of CD19 and V5/TCR α expression for cells transduced with the weak → dom TCR (red circles) and cells transduced with the dom → weak TCR (blue circles). **e** Cell surface expression levels of CD19 and V5/TCR α or CD19 and Myc/TCR β for cells transduced with the weak → dom TCR (red) and cells transduced with the dom → weak TCR (blue) as determined by flow cytometry of non-permeabilised cells

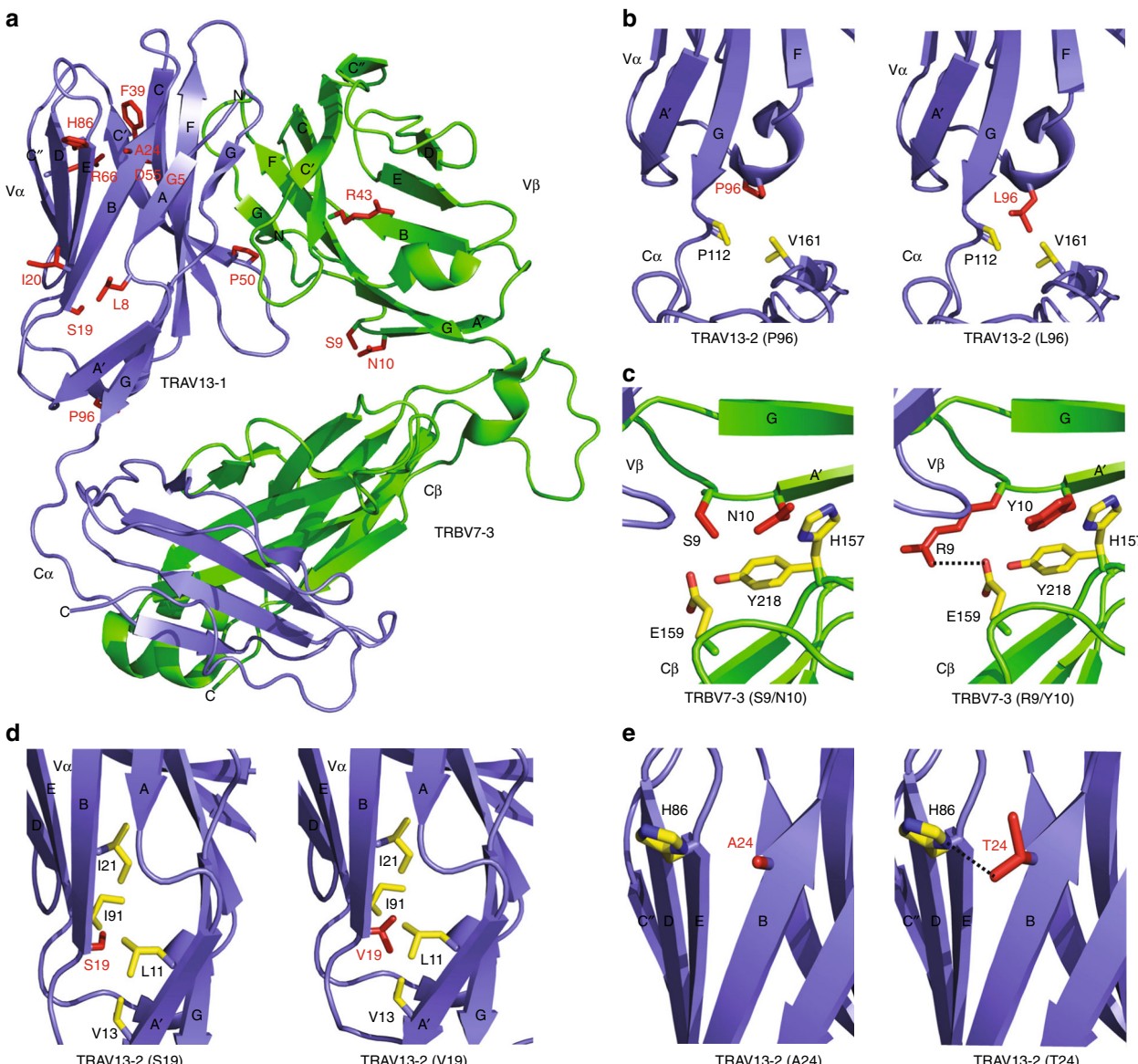

**Fig. 5** Structural modeling reveals a mechanistic role for framework residues in TCR stability. The published 3D structure of the 3PL6 TCR (TRAV13-1/TRBV7-3) was used as a model for the weak 1 TCR (TRAV13-2/TRBV7-3). **a** The location of each of the 14 residues that were changed in the weak 1 TCR to modify surface expression. **b** The change from P96α to L96α improves the interaction between the variable and constant domains of the α chain. Left panel: proline at position 96. Right panel: leucine at position 96. **c** The change from S9β and N10β to R9β and Y10β improves the interaction between the variable and constant domains of the β chain. Left panel: serine and asparagine at positions 9 and 10, respectively. Right panel: arginine and tyrosine at positions 9 and 10, respectively. **d** The change from S19α to V19α improves hydrophobic interactions within a hydrophobic core in the α chain. Left panel: serine at position 19. Right panel: valine at position 19. **e** The change from A24α to T24α can improve hydrogen bond interactions with the imidazole ring nitrogen of H86α. Left panel: alanine at position 24. Right panel: threonine at position 24. Vα variable alpha, Vβ variable beta, Cα constant alpha, Cβ constant beta

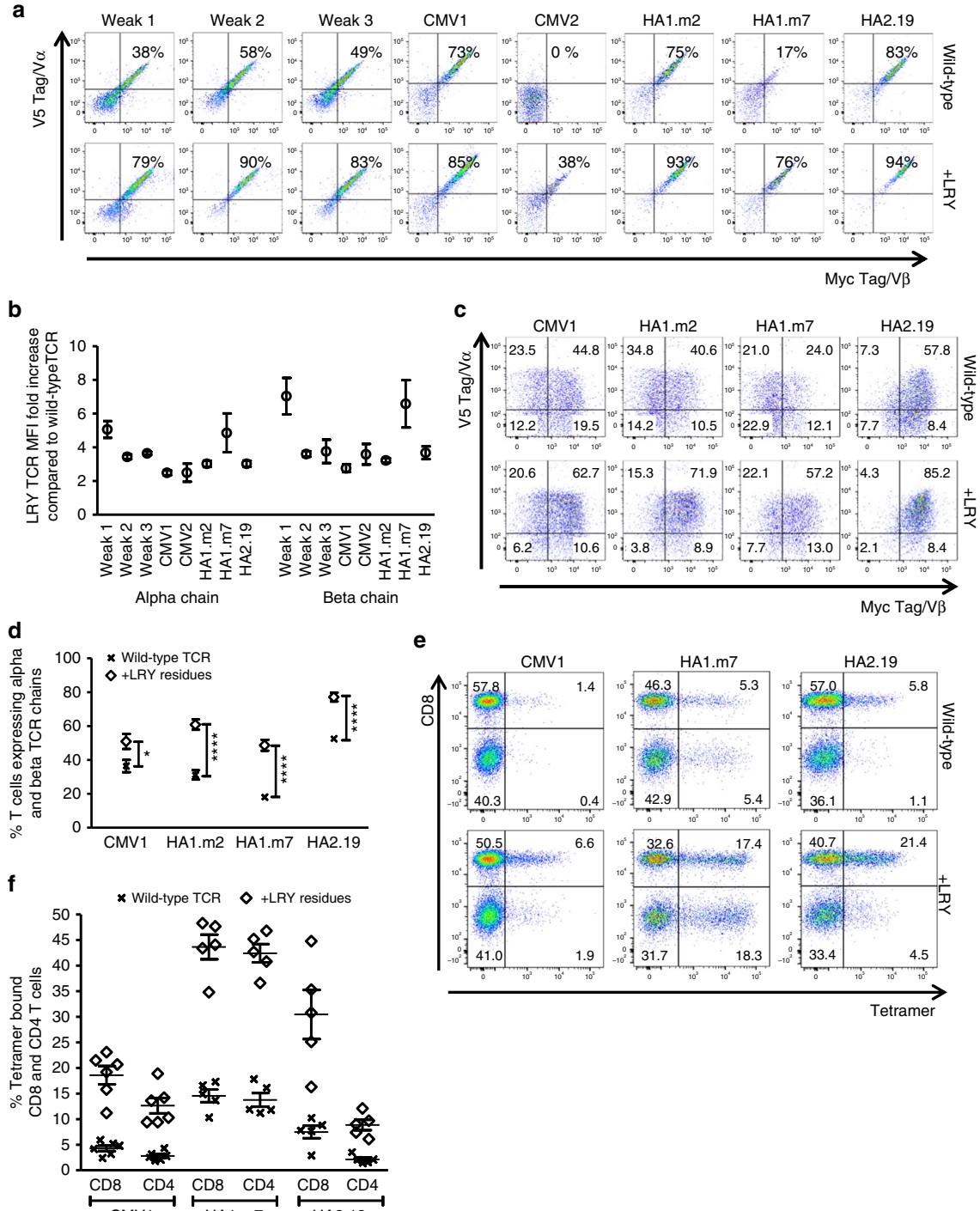

**Fig. 6** Replacement of three framework residues reduces TCR mispairing and enhances TCR expression. The roles of L96α, R9β and Y10β (LRY) were tested in three weak TCRs selected from the weak TCR library and in five antigen-specific TCRs (2 TCRs specific for CMVpp65, 2 TCRs specific for HA1[40], and 1 TCR specific for HA2). **a** A representative example of $n = 3$ independent experiments showing Jurkat cells (expressing an endogenous TCR) transduced with the indicated wild-type TCRs (top row) or the corresponding LRY-modified TCRs containing L96α, R9β and Y10β (bottom row).TCR α/β surface expression was determined on gated cells expressing equivalent levels of CD19. **b** Pooled data (means ± SEM) showing the fold increase in TCR α and β chain expression for eight LRY-modified TCRs compared with the corresponding wild-type TCRs. $n = 3$ independent experiments. MFI median fluorescence intensity. **c** Human peripheral blood T cells were transduced with the indicated wild-type or LRY-modified TCRs. The dot plots show expression of the introduced TCR α and β chains on T cells gated for equivalent expression of CD19. Data are representative of $n = 5$ independent experiments. **d** Pooled data (means ± SEM) showing the percentage of T cells expressing both TCR α and β chains when transduced with the indicated wild-type or LRY-modified TCRs. $n = 5$ independent experiments. $P$ values were less than 0.05 for all comparisons between the modified TCRs and the corresponding wild-type TCRs (unpaired $t$-test; *$P < 0.05$; ****$P < 0.0001$). **e** Shown is the percentage of tetramer-binding human T cells transduced with the indicated wild-type or LRY-modified TCRs. The data is representative of at least five independent experiments. **f** The percentages of tetramer-binding CD8[+] and CD4[+] T cells from $n = 6$ independent experiments with the CMV1 TCR, $n = 5$ independent experiments with the HA1.m7 TCR, and $n = 5$ independent experiments with the HA2.19 TCR. Data are shown as mean ± SEM. Vα variable alpha, Vβ variable beta

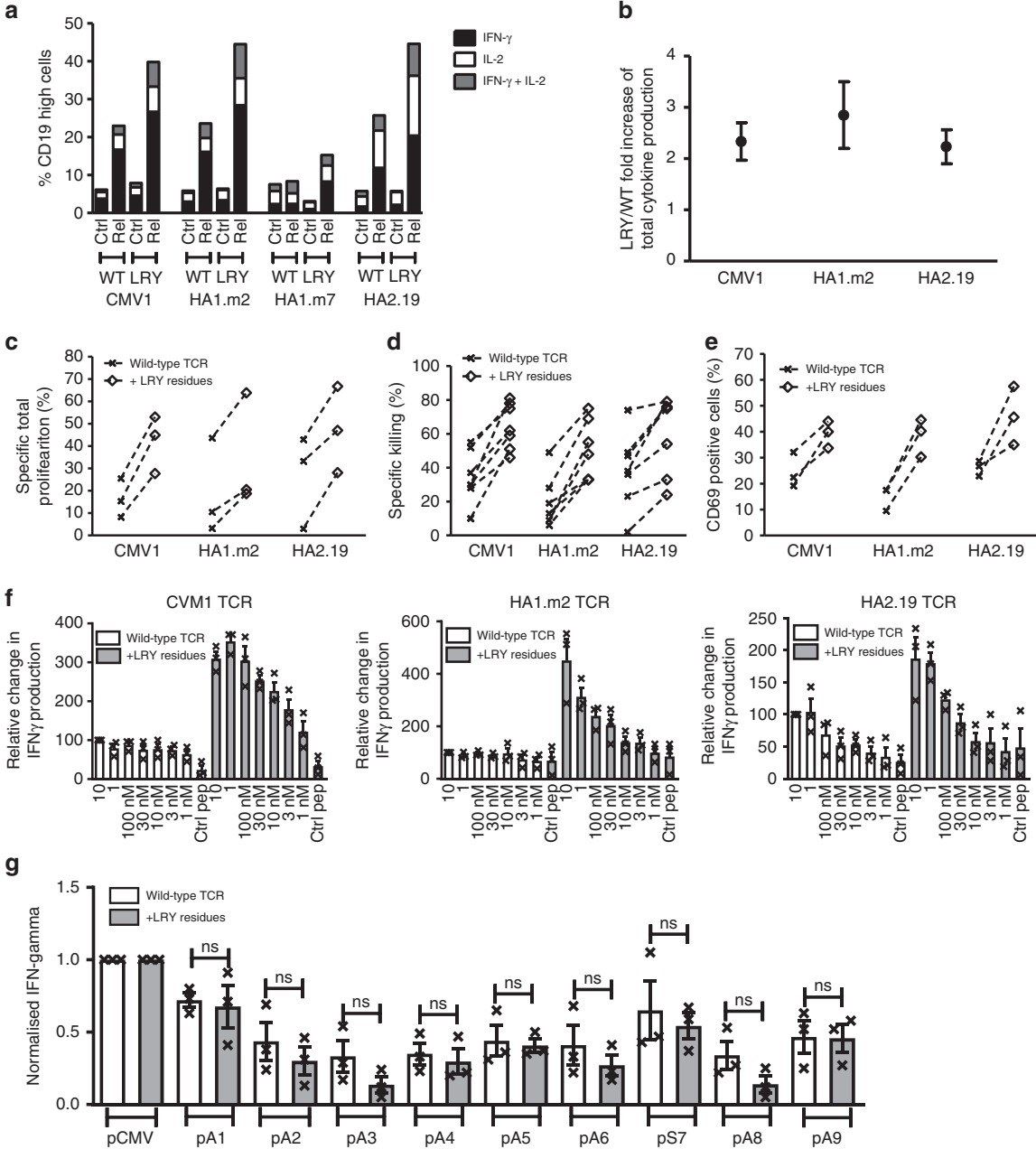

**Fig. 7** Residues L96α, R9β, and Y10β enhance antigen-specific effector functions. Human T cells transduced with wild-type or L96α, R9β, and Y10β (LRY)-modified TCRs were stimulated with peptide-loaded T2 cells. **a** A representative example of $n = 3$ independent experiments showing the frequencies of gated CD19[high] T cells that produced IFNγ and/or IL2. **b** Pooled data (means ± SEM) showing the fold increase in total specific cytokine production by LRY-modified TCR-transduced cells over the corresponding wild-type TCR-transduced cells. $n = 3$ independent experiments. Cytokine produced by stimulation with the irrelevant peptide were subtracted from cytokine produced by cognate peptide. **c** Transduced T cells labeled with Cell Trace Violet were co-cultured for 5 days with peptide-loaded T2 cells. Shown are the percentages of wild-type or LRY-modified TCR-transduced CD8[+] T cells that underwent antigen-specific proliferation. Proliferation arising from irrelevant peptide stimulation was subtracted from cognate peptide-induced proliferation. $n = 3$ independent experiments. **d** The indicated transduced T cells were co-cultured overnight with T2 cells pulsed with control peptide or cognate peptide. Shown is the antigen-specific killing of $n = 7$ independent experiments for the CMV1 TCR and $n = 6$ independent experiments for the HA1.m2 and HA2.19 TCRs. **e** Transduced T cells were co-cultured for 4 h with peptide-loaded T2 cells. Shown are $n = 3$ independent experiments measuring antigen-specific upregulation of CD69 on CD8[+] T cells expressing wild-type or LRY-modified TCRs. CD69 expression in response to irrelevant peptide stimulation was subtracted from cognate peptide induced CD69 expression. **f** Transduced T cells were stimulated overnight with T2 cells loaded with the indicated concentrations of cognate peptide. IFNγ production was measured by ELISA. Data were pooled (means ± SEM) and normalized to IFNγ production by wild-type TCR-transduced T cells stimulated with T2 cells loaded with 10 μM cognate peptide. $n = 3$ independent experiments. **g** CMV1 TCR-transduced T cells were stimulated overnight with T2 cells expressing variant or cognate peptide. IFNγ production was measured by ELISA. Data were pooled (means ± SEM) and normalized to IFNγ produced in response to cognate peptide stimulation. $n = 3$ independent experiments. ns (non-significant), $P > 0.05$ (Mann–Whitney $U$ test) for all comparisons between the LRY-modified TCR and the wild-type TCR

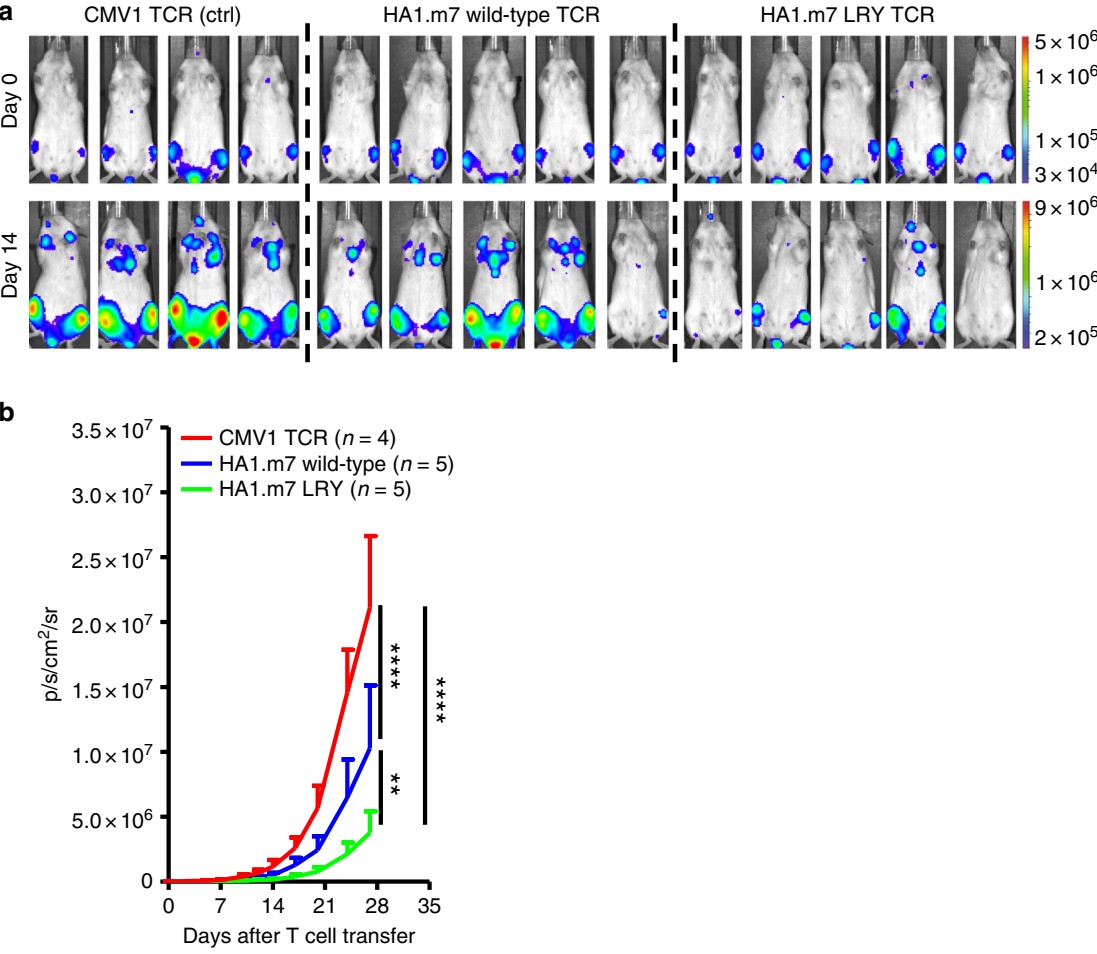

**Fig. 8** Residues L96α, R9β, and Y10β improve tumor control in vivo. NSG mice were injected i.v. with $2 \times 10^6$ HLA-A*0201$^+$ U266 multiple myeloma cells, which naturally express HA1. These cells were transduced with luciferase to enable bioluminescent imaging of the resulting tumors. **a** Top panel: all mice displayed similar tumor burdens after 10 days (day 0 of treatment). Mice were then injected i.v. with $3 \times 10^6$ human CD19$^+$ CD8$^+$ T cells expressing either the control (ctrl) CMV1 TCR ($n = 4$), the wild-type HA1.m7 WT TCR ($n = 5$) or the LRY-modified HA1.m7 TCR ($n = 5$). Bottom panel: mice injected with the LRY-modified HA1.m7 TCR showed the lowest tumor burdens on day 14 of treatment. **b** Pooled summary data (means ± SEM) of bioluminescent imaging performed at the indicated time points over a period of 28 days. Significance was determined using a two-way ANOVA with Tukey's multiple comparison test. **$P < 0.01$; ****$P < 0.0001$. Red line, mice injected with ctrl TCR. Blue line, mice injected with wild-type TCR. Green line, mice injected with LRY-modified TCR

sufficient to increase TCR expression by 3–6 fold, which mirrors the difference in expression levels between dominant and weak TCRs.

We found that TCR expression levels had a profound effect on the magnitude of antigen-specific T cell responses. In some cases, a three-fold increase in TCR density resulted in more than 1000-fold reduction in the peptide concentration that was required to trigger robust T cell effector function. This underscores the importance of the described framework region modification strategy for TCR gene therapy. We anticipate that our approach will enhance expression of most human TCRs and augment the therapeutic efficacy of gene-modified T cells. The observation that TCR density improves T cell avidity is consistent with previous studies showing that high TCR expression levels were essential for efficient T cell activation by weak agonist peptides[23]. The requirement for high TCR expression was overcome by stimulation with strong agonist peptides, indicating that enhanced TCR–peptide affinity could compensate for suboptimal TCR densities. Although many studies have explored the role of TCR affinity in T cell activation, it has been technically more challenging to modulate TCR density to determine its effect on T cell avidity[24]. Our analysis of variant peptides indicated that the LRY modification does not change the cross-reactivity profile of TCRs. However, it is important to note that enhanced antigen sensitivity allows LRY-modified TCRs to recognize lower concentrations of variant peptides, as well as lower concentrations of the cognate peptide. Clinical trials will be therefore required to assess the safety of LRY-modified TCRs and assess whether the recognition of low concentrations of variant peptides might increase the toxicity profile of adoptive therapy with T cells expressing LRY-modified TCRs.

We have shown that the LRY modification provides a TCR intrinsic benefit that leads to improved expression in cells without endogenous TCR chains. We also found that the modification is effective in reducing TCR mis-pairing in cells that do express endogenous TCRs. At present, we do not know whether the enhanced antigen-specific functional activity of LRY-modified TCRs is primarily related to the improved cell surface expression or the reduction of mis-pairing with endogenous TCR chains. We have started to address this question using CRISPR-mediated deletion of endogenous TCR α and β chains, which allows us to compare the expression and function wild-type and LRY-modified TCRs in the absence of mis-pairing with endogenous TCR chains.

How peptide binding initiates TCR signaling is not fully understood, but proposed mechanisms include serial triggering, kinetic proofreading, kinetic segregation, and conformational change[11,25–27]. A recent study comparing various TCR-signaling models concluded that a combination of kinetic proofreading and serial triggering most accurately fitted experimental data of T cell activation[28]. In this model, TCR binding to cognate peptide–HLA complexes needs to persist for a certain amount of time to achieve a signaling competent state. Although the molecular details of this signaling competent state are not clear, it is thought to include tyrosine phosphorylation by Lck, ZAP70 binding and clustering of TCRs at the center of the immunological synapse. A higher density of TCR molecules in the T cell membrane is likely to increase the probability of encountering rare cognate peptide–HLA complexes in the membrane of antigen-presenting cells, which drives synapse formation and facilitates the rebinding of clustered TCRs to peptide–HLA complexes. The three residues identified in our study stabilized the interface between the variable and constant domains, which may increase the rigidity of TCR molecules. Although it is currently not known whether TCR rigidity has any impact on the likelihood of achieving a signaling-competent state, it is a potential explanation for our observation that LRY-modified TCRs mediated vastly improved the efficiency of T cell activation at low antigen concentrations.

The framework engineering approach described here reduced the level of mis-pairing with endogenous TCR chains, which can produce novel α/β combinations of random specificity, including potential autoreactivity. Murine model experiments have clearly demonstrated that mis-pairing can cause fatal toxicity after adoptive transfer of TCR-gene-modified syngeneic T cells[29]. Similarly, in vitro studies of TCR-gene-modified human T cells revealed that mis-pairing can produce potentialy harmful novel specificities[30]. Although a causal link between mis-pairing and severe toxicity has not yet been demonstrated in patients, technologies that reduce mis-pairing are likely to mitigate the risk of unwanted toxicity in vivo.

In conclusion, the framework engineering platform described in this study provides exciting opportunities to optimize the surface expression and boost the therapeutic efficacy of human TCRs.

## Methods

**Cells, media, antibodies, tetramers, peptides, and enzymes.** Jurkat cells lacking endogenous TCR expression were obtained from Dr. F. Falkenburg, Leiden University Medical Center, Netherlands. Phoenix amphotropic packaging cells were obtained from Dr. G. Nolan, Stanford University, USA (ATCC CRL-3213). HLA-A2+ T2 cells, which lack the transporter associated with antigen processing and can be efficiently loaded with exogenous peptides, were obtained directly from the ATCC (CRL-1992). All cell lines used were routinely tested to exclude infection with *Myoplasma*. HLA-A2+ PBMCs were obtained from volunteer donors via the National Health Blood Transfusion Service (NHSBT) after obtaining NHSBT approval NCI0287/P772 and UCL Research Ethics approval (ID 15887/001) Unless otherwise stated, cells were cultured in RPMI medium (Lonza) supplemented with 10% fetal calf serum, 1% penicillin/streptomycin and 1% L-glutamine (Gibco). Phoenix amphotropic packaging cells were cultured in IMDM medium (Lonza) supplemented with 10% fetal calf serum, 1% penicillin/streptomycin, and 1% L-glutamine (Gibco). The following anti-human antibodies were used in flow cytometry experiments: anti-CD3–FITC (clone HIT3a), anti-CD3–PerCP-Cy5.5 (clone SK7), anti-CD8–APC-Cy7 (clone SK1), anti-CD69–APC (clone FN50), anti-TCR β–PerCP-Cy5.5, anti-IFNγ–FITC (clone B27), and anti-IL-2–PE (clone MQ1-17H12) (all from BD Biosciences). The following anti-murine antibodies were used in flow cytometry experiments: anti-TCR β–APC (clone H57-597; BD Biosciences) and anti-CD19–PE-Cy7 and anti-CD19–PerCP-Cy5.5 (clone ID3; eBioscience). Other antibodies used in this study were anti-V5–PE and anti-V5–APC (rabbit polyclonal; Abcam), myc purified (clone 9E10; AbD Serotec), and anti-IgG1–PE (clone A85-1; BD Biosciences). The following antibodies were used in confocal microscopy experiments: anti-murine CD19 (clone ID3; eBioscience), purified anti-V5 (goat polyclonal; Abcam), donkey anti-rat IgG1–AF488 (polyclonal; Invitrogen) and donkey anti-goat IgG1–AF546 (polyclonal; Invitrogen). PE-labeled HLA-A2/NLV, HLA-A2/HA1, and HLA-A2/HA2 tetramers were obtained from MBL. The pCMVpp65 (NLVPMVATV) and control pWT235 (CMTWNQMNL) peptides

were synthesized by ProImmune, and the pHA1 (VLHDDLLEA) and pHA2 (YIGEVLVSV) peptides were sythesized by the Core Facility at Leiden University Medical Center. The restriction enzymes Not1, SacII, NcoI, BglII, and BsrG1 were purchased from New England Biolabs.

**Generation of retroviral TCR constructs.** DNA constructs were cloned into retroviral pMP71 vectors using Not1 at the 5′ end and BsrG1 at the 3′ end. The synthetic dominant TCRs, specific for CMVpp65, Epstein-Barr virus (EBV) LMP2, or Wilm's tumor antigen 1 (WT1), were engineered previously to incorporate codon-optimized human variable domains and codon-optimized murine constant domains[31]. An extra inter-chain disulfide bond was introduced between the murine constant domains (C48α and C79β). Each gene construct incorporated a TCR α chain, a viral P2A sequence, a TCR β chain, a viral T2A sequence, and truncated murine CD19. A V5 tag was present at the N terminus of the TCR α variable domain. Two myc tags were present at the N terminus of the TCR β variable domain. The dominant TCR construct incorporating non-codon-optimized human variable and constant domains was synthesized by GeneArt (Thermo Fisher Scientific) and was designed so that the V5/TCR α variable and myc/TCR β variable domains were delimited by unique restriction sites (Not1/SacII and Nco1/BglII, respectively). The additional V5/TCR α variable segments and myc/TCR β variable segments were also synthesized by GeneArt. The remaining TCR constructs were engineered by switching the variable domains using restriction enzyme digests and ligating with the Quick Ligation (New England Biolabs). The dominant TCR expressed TRAV38-2/TRBV7-8, the weak 1 TCR expressed TRAV13-2/TRBV7-3, the weak 2 TCR expressed TRAV23/TRBV7-9, the weak 3 TCR expressed TRAV29/TRBV2, and the CMV1 TCR expressed TRAV24/TRBV6-5. The α and β variable sequences of the CMV2 TCR (TRAV12-3/TRBV20-1), the HA1.m2 TCR (TRAV13-1/TRBV7-9), the HA1.m7 TCR (TRAV25/TRBV7-9), and the HA2.19 TCR (TRAV20/TRBV18) were described previously[18,32,33]. The α and β variable segments were designated according to the IMGT nomenclature. Amino acid substitutions were introduced using either a Quikchange II XL Site-Directed Mutagenesis Kit (Agilent Technologies) or a GeneArt Site-Directed Mutagenesis PLUS Kit (Thermo Fisher Scientific).

**Expression of retroviral TCR constructs.** For retroviral production, $2 \times 10^6$ phoenix amphotropic packaging cells were cultured in 10-cm culture plates for 24 h in complete IMDM media. The cells underwent a 100% media change and were transiently transfected with the retroviral vectors (2.6 µg) and amphotropic envelope (1.5 µg) using FuGENE® HD transfection reagent (Promega). Viral supernatants were harvested 48 h following transfection. Jurkat cells were split 24 h before retroviral transduction, and PBMCs were activated for 48 h using CD3/CD28 antibody-coated Dynabeads (Thermofisher) and IL2 (Roche). Retroviral transductions were conducted on retronectin (Takara) coated 24-well plates. 500 µL of virus supernate and $1 \times 10^6$ cells were added per well, and spun at 2000 rpm, 32 °C, for 2 h. Viral supernate was removed and replaced with fresh media. Codon-optimized WT1 TCR containing murine constant domains with an additional disulfide bond were also stably transduced into Jurkat cells (referred to as Jurkat cells expressing an endogenous TCR). TCR expression on the cell surface was determined 72 h after transduction via flow cytometry. Data were acquired using an LSRFortessa (BD Biosciences) and analyzed with FlowJo software (Tree Star Inc.). For transduced Jurkat cells, single, live cells were gated for high or intermediate expression of CD19 (to normalize for transduction efficiency), and TCR expression was determined by staining for the V5/TCR α chain and the myc/TCR β chain. For transduced primary T cells, live, single cells were gated on CD3 and either CD8 and/or CD19 as appropriate, and TCR expression was determined by staining for the V5/TCR α chain and the myc/TCR β chain. For tetramer-binding studies, transduced primary T cells were gated on live, single cells, and tetramer expression determined in CD3+/CD8+ T cells.

**TCR α/β mRNA assay.** Jurkat cells were transduced with the dominant TCR, the weak 1 TCR, the dom → weak TCR or the weak → dom TCR and stained for CD19 and V5. A prime flow RNA assay (eBioscience) was then conducted using probe sets designed to bind human TCR α constant domain and human TCR β constant domain transcripts. Expression data were acquired using an LSRFortessa and analyzed with FlowJo software. Cells were gated for high expression of CD19. The α constant domain was read on AF488, and the β constant domain was read on AF647.

**Confocal microscopy.** Jurkat cells transduced with either the dom → weak TCR or the weak → dom TCR were stained with purified anti-CD19, washed, resuspended in ice-cold methanol (8 min at –20 °C), washed again, stained with donkey anti-rat IgG1–AF488, washed one more time and sorted by flow cytometry, gating for high expression of CD19. Cells were stained with purified anti-V5, washed, stained with donkey anti-goat IgG1–AF546, and cytospun onto slides (100,000 cells in 100 µl). Confocal data were collected using an inverted Nikon Eclipse Ti equipped with an ×60 oil immersion objective. Constant laser powers and acquisition parameters were maintained throughout. Digital images were prepared using Fiji.

**Antigen-specific cytokine production assays**. For intracellular cytokine production, $3 \times 10^5$-irradiated (80 Gy) T2 cells were loaded for 2 h with 10 μM peptide and co-cultured for 18 h with $3 \times 10^5$ TCR-transduced human T cells in the presence of 1 μg/ml brefeldin A (Sigma-Aldrich) in a total volume of 250 μl of culture medium per well in round-bottom 96-well plates. Cells were then surface stained for CD8 and CD19, fixed/permeabilized using a Fixation/Permeabilization Solution Kit (BD Biosciences), and stained intracellularly for IFNγ and IL-2. Data were acquired using an LSRFortessa and analyzed with FlowJo software. For extracellular cytokine secretion, $1 \times 10^5$-irradiated T2 cells were loaded for 2 h with the indicated concentrations of peptide and co-cultured for 18 h with $1 \times 10^5$ TCR-transduced human T cells in round-bottom 96-well plates containing 250 μl of culture medium per well. Supernatants were harvested from duplicate wells and tested for IFNγ and IL-2 using human ELISA Kits (BD Biosciences). Absorbance was read at 450 nm. A similar experimental set-up was used for the LEGENDplex assay (BioLegend), with the exception that irradiated T2 cells were loaded with 10 μM peptide, and supernatants were tested for secreted cytokines using a Human Th Cytokine Panel (BioLegend).

**Proliferation assay**. $2.5 \times 10^4$-irradiated T2 cells loaded with 10 μM peptide and $5 \times 10^4$ bulk-transduced T cells labeled with Cell Trace Violet (Invitrogen) were co-cultured for 5 days in round-bottom 96-well plates containing 250 μl of culture medium per well. Cells from duplicate wells were pooled, stained for CD8 and CD19, and analyzed by flow cytometry. Data were acquired using an LSRFortessa and analyzed with FlowJo software.

**CD69 upregulation assay**. $3 \times 10^5$-irradiated T2 cells loaded with 10 μM peptide and $3 \times 10^5$ bulk-transduced T cells were co-cultured for 4 h in round-bottom 96-well plates containing 250 μl of culture medium per well. Cells were stained for CD8, CD19, and CD69, and analzyed by flow cytometry. Data were acquired using an LSRFortessa and analyzed with FlowJo software.

**Antigen-specific killing assay**. Transduced T cells were expanded by stimulation with cognate peptide for 1 week prior to assay set-up. For this, $5 \times 10^5$-transduced cells, $2 \times 10^5$-irradiated T2 cells loaded for 2 h with cognate peptide, and $2 \times 10^6$-irradiated autologous PBMCs as feeder cells were co-cultured in 24-well plates in 2 mL of complete RPMI media supplemented with 10 U/mL IL-2 (Roche) for the antigen-specific killing assay, T2 cells loaded with cognate peptide and labeled with 0.02 mM CFSE were mixed at a 1:1 ratio with T2 cells loaded with control peptide and labeled with 0.2 mM CFSE. Mixed T2 cells were co-cultured with expanded transduced T cells at E:T ratios of 1:1 or less for 18 h. Antigen-specific killing was calculated from the flow cytometry data using the following equation: % specific killing = 100−[(relevant/irrelevant) with T cells/(relevant/irrelevant) with no T cells × 100].

**Peptide variant assay**. $1 \times 10^5$ T2 cells loaded with 10 μM cognate peptide or alanine (or serine) peptide variants were co-cultured for 18 h with $1 \times 10^5$ bulk-transduced T cells in round-bottom 96-well plates containing 250 μl of culture medium per well. All conditions were assayed in duplicate. Supernatants were tested for secreted IFNγ using a human IFNγ ELISA Kit (BD Biosciences) and absorbance was read at 450 nm as described above. NLV peptide variants were ALVPMVATV, NANPMVATV, NLAPMVATV, NLVAMVATV, NLVPAVATV, NLVPMAATV, NLVPMVAAV, NLVPMVATA, and NLVPMVSTV. HA1 peptide variants were ALHDDLLEA, VAHDDLLEA, VLADDLLEA, VLHADLLEA, VLHDALLEA, VLHDDALEA, VLHDDLAEA, VLHDDLLAA, and VLHDDLLES. HA2 peptide variants were AIGEVLVSV, YAGEVLVSV, YIAEVLVSV, YIGAVLVSV, YIGEALVSV, YIGEVAVSV, YIGEVLASV, YIGEVLVAV, and YIGEVLVSA. Variant peptides were synthesized by Severn Biotech Ltd.

**In vivo anti-tumor efficacy**. NOD.Cg-Prkdc(scid)Il2rg(tm1Wjl)/SzJ (NOD *scid* gamma, NSG) mice were purchased from The Jackson Laboratory and subsequently bred and maintained at the Leiden University Medical Centre Animal Facility. All mouse studies were performed in accordance with guidelines of Lieden University Medical Center after obtaining permission from the national and local Ethical Committees for Animal Research (AVD116002017891) and in accordance with Dutch laws on animal experiments. All mice were provided with water and food ad libitum. Male NSG mice were injected i.v. with $2 \times 10^6$ U266 multiple myeloma cells transduced with luciferase (pCDH-EF1-Luc2-P2A-tdTomatoRed), obtained from Kauhiro Oka via Addgene (plasmid 72486). For tumor visualization, mice were injected i.p. with 200 μl of 7.5 mM D-luciferine (Cayman Chemical Co.) and anesthetized with 3% isoflurane. Bioluminescent images were obtained using a CCD camera (IVIS Spectrum, PerkinElmer). Ten days after tumor outgrowth, mice were injected i.v. with $3 \times 10^6$ CD19+ CD8+ T cells expressing the wild-type HA1. m7 TCR, the LRY-modified HA1.m7 TCR or the control CMV1 TCR. Tumor growth was monitored at 2–3-day intervals for a total of 28 days. Statistical analysis was performed using Prism software version 7 (GraphPad). Significance was determined using a two-way ANOVA with Tukey's multiple comparison test.

**Sequencing of endogenous dominant and weak human TCRs**. Peripheral blood T cells from healthy donors were transduced with retroviral vectors encoding strongly expressed synthetic TCRs containing human variable domains and murine constant domains with an artificial disulfide bond between residue 48 of the α chain and residue 79 of the β chain. Antibodies specific for the human constant β domain were used to assess expression of endogenous human TCRs, and antibodies specific for the murine constant β domain were used to assess expression of the introduced synthetic TCRs. Flow cytometric sorting was used to purify transduced donor T cells that either co-expressed the endogenous and introduced TCRs or expressed only the introduced TCR. Two methods were used to determine the endogenous TCR sequences from extracted mRNA. The first method employed an unbiased template-switch anchored RT-PCT to amplify all expressed TCR alpha and TCR beta gene rearrangements, which were then subcloned and sequenced using a conventional Sanger-based approach[34,35]. The second method employed total RNA isolated from sorted T cells, collected in Tempus™ Blood RNA tubes (Thermofisher #4342792) using the manufacturer's protocol for RNA extraction. The method introduces unique molecular identifiers attached to individual cDNA molecules to provide a quantitative and reproducible method of library preparation. Full details for both the experimental TCRseq library preparation and the subsequent computational analysis (V, J, and CDR3 annotation) using Decombinator was recently published[21,36].

**Statistical TCR analysis**. TCR library generated by Sanger sequencing: the nucleic acid sequences were translated into amino acids and aligned using IMGT reference numbering (http://www.imgt.org/). To compare amino acid frequencies at every residue in the dominant and weak TCR libraries, $2 \times 2$ contingency tables were computed for each position in the alignment, representing the observed counts of each amino acid type A versus all other types (~A) in the dominant versus the weak TCR sequences. The null hypothesis was that the relative frequency of occurrence of amino acid type A was the same in the dominant and the weak TCR sequences at a given position, and $P$ values were calculated from the hypergeometric distribution using Fisher's exact test without correction for multiple comparisons. Rejection of the null hypothesis indicated that amino acid type A was significantly enriched in either the dominant or the weak TCR sequence at a given position.

TCR library generated by next generation sequencing: at each position of the framework regions we compared the frequency of all amino acids in the dominant and in the weak TCR libraries using either Fisher's exact test performed on each donor independently, or the Cochran–Mantel–Haenszel test (CMH) performed on both donors together. The results were adjusted for multiple comparisons using the Bonferroni correction.

**TCR 3D structural modeling**. The weak TCR that was most extensively tested in our study comprised TRAV13-2, TRBV7-3. The TRAV13-2 chain was modeled with I-TASSER (Iterative Threading ASSEmbly Refinement) server[37] using the closely related TRAV13-1 structure (PDB code 3PL6) as a threading template. The 3PL6 TCR structure consists of the TRAV13-1 chain paired with TRBV7-3, the same chain that is present in our weak TCR. Therefore, the I-TASSER-derived TRAV13-2 model was superimposed onto the TRAV13-1 chain of the 3PL6 structure to generate a molecular model of TRAV13-2/TRBV7-3 TCR complex. Models of weak to strong TCRs incorporating the 14 variable domain framework residues were also generated using the I-TASSER server. For all modeling studies with I-TASSER, the target sequences were initially threaded through the PDB library by the meta threading server, LOMETS2. Continuous fragments were excised from LOMETS2 alignments and structurally reassembled by replica-exchange Monte Carlo simulations. The simulation trajectories were subsequently clustered and used as the preliminary state for second round I-TASSER assembly simulations. Finally, lowest energy structural models were selected and refined by fragment-guided molecular dynamic simulations to optimize hydrogen-bonding interactions and remove steric clashes. Analysis of molecular interactions was performed using programs of the CCP4 suite[38]. Model visualization was carried out using COOT[39]. Structural figures were generated using PyMOL (The PyMOL Molecular Graphics System, Version 1.8 Schrödinger, LLC).

## Data availability
The data generated during and/or analyzed during the current study are available from the corresponding author on reasonable request. TCR sequences have been deposited to BioProject database under the accession number SUB6223064.

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

## Acknowledgements

This work was supported program grant 13004 from Bloodwise and by a research grant from Cell Medica. Additional support was obtained from the Medical Research Council, CRUK Experimental Cancer Medicine Centre, NIHR UCL/UCLH Biomedical Research Centre and Wellcome Trust. D.A.P. is supported by a Wellcome Trust Investigator Award (Grant code 100326/Z/12/Z). B.E.W. and F.M. are supported by a Wellcome Trust Investigator award to B.E.W. (Grant code 099266/Z/12/Z). R.M.R. is supported by the PPP Allowance made available by Health Holland, Top Sector Life Sciences and Health. We thank Emma Gostick and Kristin Ladell for technical assistance.

## Author contributions

S.T. designed and conducted experiments, analyzed data and wrote the paper; R.M.R., A.W., A.K., F.M., T.S., A.K., D.S., A.H., L.G., D.J., K.K.M. conducted experiments and analyzed data; E.M. designed experiments, B.M.C., D.A.P., M.H.M.H., B.E.W. designed experiments and wrote the paper, H.J.S. initiated the study, designed experiments, analyzed data, and wrote the paper.

## Competing interests

H.J.S. is share holder of Cell Medica and obtained research funding from Cell Medica and Apollo Ltd. H.J.S. and E.M. are founders, consultants, and share holders of Quell Therapeutics. S.T., F.M., B.E.W., and H.J.S. are inventors of a patent of the dominant TCR technology. The remaining authors declare no competing interests.
