## [Peer Review File · Nature Communications]

Reviewers' comments:

Reviewer #1 (Remarks to the Author):

Thomas et al. identified 14 amino acid residues in the framework of the TCR that affect transgenic surface expression. By replacing these residues, the authors can enhance surface expression of weak recombinant TCRs and in turn deteriorate surface expression of strong recombinant TCRs. Enhanced surface expression seems to come along with enhanced in vitro functionality.

The generation of defined T cell products with predictable efficacy is highly relevant for the field of T cell therapy, and modulation of TCR surface expression is an interesting approach to enhance functionality of T cells. The authors nicely demonstrate skewed distributions of different motifs in TRAV and TRBV in strong and weak TCRs and validated important motifs by TCR re-expression.

Although the work is well performed and should be of significant interest to the field of T cell biology and T cell engineering, there are some general aspects that should be addressed (see below).

Importantly, for clinical translation safety is of utmost importance and the provided analyses on potential changes in the fine specificity of modified receptors are not sufficient/convincing. It would help to show alanine peptide scans for several different unmodified/modified TCRs and to perform more intensive statistical analyses on these data. Furthermore, the authors should present and discuss this part much more carefully in order to give the reader a clear understanding that the "safety" aspect of TCR framework engineering for therapeutic applications still need to be addressed in future studies (especially in in vivo experiments and first-in-man clinical trials).

Major points:

1. Re-expression of transgenic TCRs leads to a spectrum of cells showing either transgenic expression alone, dual expression of transgenic receptor and endogenous receptor, or mispaired variants (Fig. 1B). Does the sequencing in Fig. 1D exclude a specific influence of the particular transgenic receptor used (the exact sequencing methodology should also be described in more detail in the manuscript - see also below). Wouldn't it be necessary to perform similar analyses with several different transgenic TCRs to allow for more generalized statements?

The authors do investigate mispairing in Fig. 6 later on, but it is overall not clear if the identified 14 amino acid residues primarily render TCRs more or less prone to mispairing or if they enhance or deteriorate receptor surface expression in a TCR-intrinsic manner. In Jurkat cells TCR expression is being investigated with or without the role of the endogenous TCR, but this only allows interpretation for potential mispairing with the one Jurkat TCR. The authors should therefore investigate the expression of weak and dominant TCRs with substitution of key amino acid residues in (polyclonal) primary human T cells with or without complete knockout of the endogenous TCR.

2. The authors generated a substantial library of "dominant" and "weak" TCR sequences (Fig. 1; from how many donors? such more specific information is largely missing) and in the end tested LRY residue exchange in a significant number of TCRs (Fig. 6). While it is understandable that not all residue combinations can be tested on a large number of TCRs in different ways, very generalized statements are being made in the manuscript on the basis of potentially very unique cases. The identified biggest differences through individual residues in Fig. 3B are based on the residue exchange performed with a single TCR (the "bioinformatic analysis" should also be explained more specifically in the main text). 3D modelling revealed that it is exactly the residues at the V-C interface that have the biggest impact on TCR surface expression. It is unclear if these residues would also lie at the V-C interface for other TCRs and this should also be investigated for other weak TCRs as well as strong TCRs. "Structural modelling" should also be explained more specifically in the methods section or as supplementary document. How close to the real 3D structure is the model expected to be?

3. The functional testing in Fig. 7 is not entirely convincing. For Fig. 7A-E negative and positive controls are missing as well as representative primary data. In Fig. 7F, there is no dose-dependent response for the WT forms of CMV1 and HA1.m2 TCRs. The described 3000-fold higher responsiveness therefore can only be deduced through the level of expression at certain antigen doses, but more general EC50 values would better describe overall peptide sensitivity. The authors should explain why sigmoid antigen responses cannot be determined for the WT versions and also test functionality in

primary human T cells without the endogenous TCR to exclude a bias through the endogenous TCR and investigate to what extent the gained functionality through LRY editing can be reached through endogenous TCR KO alone, and whether LRY edited TCRs would further benefit from missing competition. Furthermore, functionality should be more extensively tested, e.g. including killing assays.

4. The safety risks of editing variable regions of TCR for potential therapeutic applications are insufficiently addressed both in terms of experimental data as well as in terms of discussion in the manuscript. The results from Fig. 7G are hard to interpret in a meaningful way. While it is not even indicated whether differences between WT and LRY TCR are potentially statistically significant, it is even more difficult to predict if this result shows a limited danger of cross-reactivity through LRY residue exchange in general, since only one TCR has been investigated. Cross-reactivity studies should be extended to several other TCRs and the limitation of these experiments for safety prediction in therapeutic in vivo applications should be clearly stated.

Minor points:

1. As indicated already above, many experiments and methods should be explained in greater detail. For each figure, the number of replicates, independent experiments and the statistics should be explained more clearly in a structured manner. Also the hybridoma clones for the main antibodies used for the studies should be mentioned?

2. How was sequencing performed? What is the composition of the sequencing library (how many TCRs from how many donors and so on)? How were alpha-beta pairs of weak and strong TCRs identified (via single cell PCR?).

3. Some of the figures miss axis labelling (e.g. Fig. 1D) or would benefit from legends (e.g. Fig. 4C).

4. Fig. 4C: Can membrane-bound and cytosolic TCR signal be distinctly quantified? 5. In Suppl. 5. In Suppl. Fig. 2 it is mentioned "This is a representative of one individual experiment"; how can this be "representative"?

Reviewer #2 (Remarks to the Author):

Review for Thomas et al, Framework engineering to produce dominant T cell receptors with enhanced antigen-specific function

A poorly characterized mechanism by which T-cells are able to fine tune their sensitivity is through enhanced TCR avidity by modulating the density of TCR at the cell surface. TCR gene therapy applications looking to exploit this mechanism have focused on increasing the total surface expression and minimizing mispairing between exogenous and endogenous TCR chains through engineering of the $\alpha\beta$ interface, either the TCR transmembrane domain, CaC β interface or VaV β interface.

Thomas et al have identified a small number of TCR framework mutations which mutate the VaCa or V β C β interface. Surprisingly despite not sitting at the $\alpha\beta$ interface these mutations reduce mispairing between endogenous and exogenous TCR and increased TCR density at the T-cell surface. This translates to a substantial enhancement of antigen specific responses. TCR avidity is often overlooked in the literature which focuses primarily on TCR affinity enhancing mutations. There are two main limitations of the study. Firstly the statistical analysis used to identify TCR framework mutations is limiting and secondly additional experiments are required to support the conclusions.

The authors identify two T-cell populations which may correlate with relative high or low TCR surface density, providing a novel approach to identify candidate framework mutations which may lead to improved surface display. However this dataset is not fully exploited due to the limited statistical analysis used. From tableS1 ~70% of the α framework positions appear to be statistically significant (no data is shown for the β -chain). Differences between the two TCR populations are dominated by a

small number of v-genes highly abundant in PBMCs. It is likely that as a result the paired statistical analysis fails to highlight the subtlety of the dataset. As a result the authors have primarily restricted their mutagenesis panel by structural analysis. Whilst this is of merit it undervalues the potential strength of their previously outlined approach.

1. There is no analysis of how frequently identified mutants are found in other V-genes. Is the identified mutant due to a reversion to the V-gene consensus at this position or a novel stabilizing mutation limited to few v-genes?

For example whilst it is true that R9 β 10Y β are statistically more common in the high surface expressing TCRs, why are the 13 TRBVs with R9 β 10Y β not enriched in the dominant population? Some R9 β 10Y β V-genes (TRBV 7-7,11-3,25-1,27) appear to be enriched in the weak population!

A pairwise analysis of each V-gene dominant or weak, to the consensus V-gene may assist the current statistical model. For example poor surface expression associated with TRAV29 has been attributed to K44 which deviates from the consensus Q44 modulating the $\alpha\beta$ interface. See US patent 20180162922

2. Due to the limited sequencing depth most V-genes are poorly represented

3. TCR sequence data is unpaired, α chain sequences are not linked to their associated β -chains which significantly reduces the analysis that can be done.

4. There is no analysis of J-region usage (linked to $\alpha\beta$ interface stability)

<https://doi.org/10.1016/j.molimm.2008.09.021>

5. There is no analysis of CDR3 length and composition (linked to $\alpha\beta$ interface stability)

<https://doi.org/10.1016/j.molimm.2010.01.012>

Additional experiments that would significantly improve confidence in the various conclusions drawn are:

1. Introduced mutations lead to improved TCR stability

Validation required: The thermal stability of the TCR should be assessed (typically measured using DSC or DSF).

2. Increased TCR expression leads to substantial enhanced T cell potency

Validation required: It should not be assumed that antigen distal residues are unable to impact on the affinity of the TCR for target antigen. At least in antibodies, this would not be unprecedented (<https://doi.org/10.1073/pnas.1613231114>). The affinity of the TCR for antigen (typically measured by BIAcore) should be determined.

3. "We anticipate the V-region engineering approach we outline can enhance surface expression of most human TCRs"

Validation preferred: Although 8 TCRs have been tested greater confidence would be placed in this statement if different TCRs were selected. As an example at IMGT 96, 10 amino acids are utilised by the 47 TRAVs. However in 6/8 TCRs tested the mutant was P96aL. Whether Leu leads to improved stability when substituting other amino acids, or indeed if Leu is the best a.a at this position is unknown.

4. Introduced mutations lead to improved surface expression

Validation preferred: Whilst relative intensity MFI clearly demonstrates an improvement it would improve the study if fluorescence intensity could be quantified using an antibody specific standard curve for example using a QuantumTM Simply Cellular kit (Bangs Laboratories, Inc.)

5. TCR density improves T cell avidity

Validation preferred: Use pHLA multimers to show enhanced staining.

Below are a few minor changes that should be addressed prior to publication:

1. Introduction, line 2: "The ability of engineered T cells to be stimulated by low concentration of peptide antigen is a key parameter for the efficacy of TCR gene therapy. Hence, mutagenesis of the antigen binding regions CDR1, 2, and 3 has been employed to select TCRs with enhanced affinity for the cognate peptide antigen." This doesn't really make sense. Natural T-cells have been shown to activate against very low epitope levels (PMID: 12397360) so why do you need to enhance the affinity? I think the authors are talking about cancer specific TCRs which are often weak affinity and do not enable cancer specific T-cells to control tumours very effectively. This should be outlined and referenced (PMID: 22949370).
2. Introduction, line 6: "However, affinity matured TCRs display enhanced cross-reactivity, which can increase the risk of off-target toxicity in patients". This statement is not accurate and is very misleading. It has not been shown that affinity matured TCRs are more cross-reactive (a sweeping statement with little evidence!). In one of the examples they cite, the affinity matured induces off-target toxicity through a single self-peptide expressed in cardiac tissue. In the second example, the neurological toxicity was attributed to ON-TARGET toxicity because the target antigen was also expressed in the human brain. Additionally, there are several examples of affinity matured TCRs that do not elicit off-target effects (in the clinic and published). The authors should delete this sentence.
3. Introduction line 9: "Consequently, TCRs with un-physiologically high affinity fail to trigger T cell responses at low peptide concentrations that are sufficient for triggering robust responses by wild-type TCRs". The last part of this sentence does not make sense. I assume the authors mean compared to wild-type TCRs with μM affinities? Alternatively: "Consequently, TCRs with super-physiologically high affinity can fail to trigger T cell responses at low peptide concentrations."
4. Introduction generally: The introduction is very brief, and quite poorly written, particularly the first paragraph. The authors should consider re-writing the introduction with more consideration for the published literature.
5. The nomenclature for naming of mutants is unconventional. This is particularly problematic for the TCRs in Fig3. Dom TCR: L39a, R55a, Q43b appears to correspond to the WT TCR.
6. The TCR described as wdom TCR F39a, D55a & R43b is actually reverting 3 of the 15 mutations that have been previously introduced back to the germline encoded residues.
7. Standard nomenclature should be used throughout. Mutants should be P96aL (not P96a to L96a mutant or Position 96 alpha proline  leucine). The authors flip between TRAV12-2 and the less conventional format TRAV12.2. In some instances a and b are used instead of α and β .
8. Conclusions: The authors state: "For some TCRs analysed, a 3-fold increase in TCR density resulted in more than 1000-fold reduction in the peptide concentration that was required to trigger robust T cell effector function." What do they mean by 'a robust T cell effector function'? They do not define what this is, or show it experimentally. All they can say is that the E_{max} was higher for INF γ and IL2 release at very high peptide concentrations. The effect is reduced at more physiological peptide concentrations (1-3 nM). EC_{50} s would be more informative.
9. Conclusions: The authors state: "The observation that TCR density improves T cell avidity is consistent with previous studies showing that high TCR expression levels were essential for efficient T cell activation by weak agonist peptide." They do not observe that TCR density improves T-cell avidity. They need to show this (tetramers??) to make this claim.
10. Conclusions: "How peptide binding initiates TCR signalling is not fully understood, but proposed mechanisms include serial triggering, kinetic proofreading and conformational change". Additionally, the authors should consider how TCR density might alter triggering via the kinetic segregation model.

11. Figures:

Fig 1A: Label each domain and Cysteine position.

Fig 1B: Label non-transduced in bottom right quadrant (rather than text body)

Fig 1D: Order V-genes according to dominance rather than numerical.

Fig 1E: reformat to be consistent with the clearer SupTable1

Fig2/FigS2: Share the same Fig legend "expressing an endogenous TCR" should be in the title

Fig2C: Label CD19 high/ CD19 intermediate

Fig5: Fig legends within figures are not necessary. Identifying crystal structures of other V-genes which already include the combination of mutations that are being modelled would have assisted in the modeling approach. The labels in the figure are likely going to be too small to read in print. The authors need to remake this figure with consideration to how it will appear in print. It is poor resolution (could just be the PDF) and a bit thrown together at present.

Fig7F: Why does the CVM1 wild-type TCRs fail to generate a dose response to the different peptide concentrations? The response is very similar at all concentrations tested. This would indicate that this TCR just has a lower Emax than the LRY modified TCRs and might be sensitive to lower concs of peptide if titrated down further. The authors need to calculate EC50s for these experiments. The HA.m2 wild-type seems to produce no INF γ response to cognate peptide at any dose compared to the irrel peptide. How do the authors explain this? In figure 7A-E this TCR does generate responses to the peptide?

Additional literature that would be of benefit for the introduction and discussion would be from two main areas. In particular those focussed around the role of T cell avidity in gene therapy (reviewed here) (<https://doi.org/10.1186/1479-5876-3-35>) and other attempts to engineer the TCR in the TCR framework/ antigen distal regions such as <https://doi.org/10.1093/protein/gzq113> and <https://doi.org/10.4049/jimmunol.1303209>.

Cover Note:

We are grateful to the reviewers for their constructive comments. We have completed a substantial amount of additional experimental work to address the concerns raised by the reviewers and to add new *in vivo* data to further improve the manuscript.

- As requested by reviewer 1 we have extended the cross-reactivity studies to 3 different TCRs confirming that the LRY-modification of the variable region framework does not result in new cross-reactivity.
- As requested by reviewer 2 we have used next generation sequencing to generate a large library of dominant and weak TCRs. The statistical analyses revealed that the majority of amino acids that were over-represented in dominant TCRs in the original small library (884 clonotypes) were also over-represented in the new large library (130000 clonotypes). The analyses of large library revealed over-representation of additional residues that were not identified in the small library. More than 80% of the 77 variable region framework positions in the α and β chain showed over-representation of certain amino acids in dominant TCRs. This indicated that the combination statistical analyses and structural modeling was required to identify candidate residues for functional testing.
- We have used an immunodeficient mouse model to show that human T cells expressing LRY-modified TCRs showed more effective control of tumor growth *in vivo* than human T cells expressing wild-type TCRs.

Together, the additional data have improved the quality of the revised manuscript.

Below is a detailed point by point reply to all the issues raised by the reviewers.

Point by Point Reply to reviewers comments

Reviewer #1 (Remarks to the Author):

Thomas et al. identified 14 amino acid residues in the framework of the TCR that affect transgenic surface expression. By replacing these residues, the authors can enhance surface expression of weak recombinant TCRs and in turn deteriorate surface expression of strong recombinant TCRs. Enhanced surface expression seems to come along with enhanced *in vitro* functionality. The generation of defined T cell products with predictable efficacy is highly relevant for the field of T cell therapy, and modulation of TCR surface expression is an interesting approach to enhance functionality of T cells. The authors nicely demonstrate skewed distributions of different motifs in TRAV and TRBV in strong and weak TCRs and validated important motifs by TCR re-expression.

Although the work is well performed and should be of significant interest to the field of T cell biology and T cell engineering, there are some general aspects that should be addressed (see below). Importantly, for clinical translation safety is of utmost importance and the provided analyses on potential changes in the fine specificity of modified receptors are not sufficient/convincing. It would help to show alanine peptide scans for several different unmodified/modified TCRs and to perform more intensive statistical analyses on these data. Furthermore, the authors should present and discuss this part much more carefully in order to give the reader a clear understanding that the “safety” aspect of TCR framework engineering for therapeutic applications still need to be addressed in future studies (especially in in vivo experiments and first-in-man clinical trials).

Major points:

1. Re-expression of transgenic TCRs leads to a spectrum of cells showing either transgenic expression alone, dual expression of transgenic receptor and endogenous receptor, or mispaired variants (Fig. 1B). Does the sequencing in Fig. 1D exclude a specific influence of the particular transgenic receptor used (the exact sequencing methodology should also be described in more detail in the manuscript - see also below). Wouldn't it be necessary to perform similar analyses with several different transgenic TCRs to allow for more generalized statements?

REPLY:

We apologize for the inadvertent omission of methodological details re sequencing of TCRs. We have added this information in the Methods section on page 21 of the revised manuscript.

Initially we employed unbiased molecular analysis using a template-switch anchored RT-PCR and Sanger sequencing as described previously (J Exp Med. 2005 Nov 21;202(10):1349-61; Curr Protoc Immunol. 2011 Aug;Chapter 10:Unit10.33). This method was used to sequence 884 distinct TCR clonotypes in 3 different donors; half of the clonotypes with a dominant and half with a weak expression phenotype (see page 4 of the revised paper). We have observed the same endogenous TCR expression profile with three different ‘synthetic’ TCRs that have a murine constant region with an additional cysteine bond (Gene Ther. 2008 Apr;15(8):625-31 and Suppl Fig 1A). In these ‘synthetic’ TCRs the murine constant regions plus the additional cysteine bond are primarily responsible for strong expression and suppression of endogenous TCRs in human T cells; the variable regions play a minor role in this context. Hence, we do not think that the variable region of synthetic TCRs has an effect on which endogenous TCRs are suppressed or co-expressed.

Reviewer 2 has asked us to perform next generation sequencing to extend the comparison of ‘dominant’ and ‘weak’ endogenous human TCRs to a larger TCR library. Hence, we used next generation sequencing (see Methods section on page 21) to produce a library of 23,511 alpha and 38,920 beta clonotypes with dominant expression, and 29,419 alpha and 41,980 beta clonotypes with a weak expression phenotype. We have performed additional statistical analyses of these large TCR libraries (see page 4 and suppl. Fig 1 of the revised manuscript and reply to point 2 of reviewer 2).

The authors do investigate mispairing in Fig. 6 later on, but it is overall not clear

if the identified 14 amino acid residues primarily render TCRs more or less prone to mispairing or if they enhance or deteriorate receptor surface expression in a TCR-intrinsic manner. In Jurkat cells TCR expression is being investigated with or without the role of the endogenous TCR, but this only allows interpretation for potential mispairing with the one Jurkat TCR. The authors should therefore investigate the expression of weak and dominant TCRs with substitution of key amino acid residues in (polyclonal) primary human T cells with or without complete knockout of the endogenous TCR.

REPLY

Our described modifications improve surface expression in the absence of endogenous TCR (suppl. figures 2b and 4a,b). The modifications also reduce mis-pairing with endogenous TCRs in primary human T cells (Fig 6c,d). The following observations indicate that both improved TCR expression and reduction of mis-pairing play a role in primary T cells:

[REDACTED]

CRISPR deletion of endogenous TCR chains has two effects: it completely avoids mis-pairing and it also removes the competition between endogenous and introduced TCRs for CD3.

[REDACTED].

Dissecting the role of reduced mis-pairing versus enhanced expression of LRY-TCRs in CRISPR edited primary T cells is part of a new project that has been started and will require substantial amounts of additional work. We have added on page13 of the revised manuscript a section on how CRISPR editing will be used to dissect the contribution of reduced mis-pairing versus enhanced expression to TCR function.

2. The authors generated a substantial library of “dominant” and “weak” TCR sequences (Fig.1; from how many donors? such more specific information is largely missing) and in the end tested LRY residue exchange in a significant number of TCRs (Fig. 6). While it is understandable that not all residue combinations can be tested on a large number of TCRs in different ways, very generalized statements are being made in the manuscript on the basis of potentially very unique cases. The identified biggest differences through individual residues in Fig. 3B are based on the residue exchange performed with a single TCR (the "bioinformatic analysis" should also be explained more specifically in the main text). 3D modelling revealed that it is exactly the residues at the V-C interface that have the biggest impact on TCR surface expression. It is unclear if these residues would also lie at the V-C interface for other TCRs and this should also be investigated for other weak TCRs as well as strong TCRs. "Structural modelling" should also be explained more specifically in the methods section or as supplementary document. How close to the real 3D structure is the model expected to be?

REPLY

Please see the additional information about the TCR library above.

We have added details of the structural modelling in the Methods section on page 21/22 of the revised manuscript.

Although we did make use of single TCRs to introduce and assess the effects of individual mutations, importantly this was built on a large amount of modelling/analyses involving a wide range of TCRs in the published structural PDB database (far larger than indicated in the original manuscript). With this in mind, we therefore agree with the reviewer's suggestion that the manuscript would benefit from a comprehensive description of the modelling/structural analysis methods employed, and this will be provided in the revised manuscript, including chiefly use of iTasser modelling software, and structural programs such as CONTACTS (CCP4) for analysis of molecular interactions. Based on these analyses, which highlight the overall highly structurally conserved nature of the framework regions of the TCR, we can confidently state that the position of the residues, including the LRY residues ultimately selected for optimization of TCR expression, will be conserved across all TCRs, given their strong structural similarity. Therefore those at the V-C interface of one TCR will be in this interface in all others. Regarding how close our models are to actual 3D structures, clearly this is only assessable by directly solving TCR structures for those V-regions modelled. However, of relevance, since submission, a published structure of TRAV38-2 has been determined, and validates the conclusions regarding this V-region from our modelling analyses.

3. The functional testing in Fig. 7 is not entirely convincing. For Fig. 7A-E negative and positive controls are missing as well as representative primary data. In Fig. 7F, there is no dose-dependent response for the WT forms of CMV1 and HA1.m2 TCRs. The described 3000-fold higher responsiveness therefore can only be deduced through the level of expression at certain antigen doses, but more general EC50 values would better describe overall peptide sensitivity. The authors should explain why sigmoid antigen responses cannot be determined for the WT versions and also test functionality in primary human T cells without the endogenous TCR to exclude a bias through the endogenous TCR and investigate to what extent the gained functionality through LRY editing can be reached through endogenous TCR KO alone, and whether LRY edited TCRs would further benefit from missing competition. Furthermore, functionality should be more extensively tested, e.g. including killing assays.

REPLY

The control in Fig 7a-e was stimulation of TCR-transduced T cells with T2 cells loaded with a control peptide. We have changed Fig 7a to show that less than 10% of T cells produced cytokine when stimulated with the control peptide. Similarly, irrelevant control peptides were used in Fig 7 b-e with less than 10% of T cells showing a response. This irrelevant response was subtracted from the response seen with cognate peptide. We have added this information to the figure legend.

EC50 values may be misleading as the maximal responses of wild type TCRs are poor. For example, the response of the wild type TCRs at the highest peptide concentration is often lower than the half maximal response of the

modified TCRs. It is therefore more useful to compare the respective peptide concentration that is required to trigger a similar magnitude of response by wild-type and modified TCRs.

Please see our reply to point 1 above regarding the analysis of LRY-modified TCRs in CRISPR-edited primary T cells that do not express endogenous TCRs. In Fig 7d of the original manuscript we have used CD107A up-regulation as a surrogate marker for cytotoxicity. We have now repeated these experiments using a flow-based assay to directly measure the killing of target cells. The results of the killing assays were similar to the CD107A assay. In the revised manuscript we have replaced in Fig 7d the CD107A data with the data of the killing assays.

4. The safety risks of editing variable regions of TCR for potential therapeutic applications are insufficiently addressed both in terms of experimental data as well as in terms of discussion in the manuscript. The results from Fig. 7G are hard to interpret in a meaningful way. While it is not even indicated whether differences between WT and LRY TCR are potentially statistically significant, it is even more difficult to predict if this result shows a limited danger of cross-reactivity through LRY residue exchange in general, since only one TCR has been investigated. Cross-reactivity studies should be extended to several other TCRs and the limitation of these experiments for safety prediction in therapeutic in vivo applications should be clearly stated.

REPLY:

As suggested by the reviewer, we have analysed the cross-reactivity profiles of two additional TCRs using alanine peptide variants. We have added the data of the two additional TCRs in supplementary Fig 6. Together, the analysis of 3 TCRs showed that there was no significant difference in cross-reactivity profile of wild types TCRs compared to LRY-modified TCRs. The additional experiments are described in the result sections on page 10 of the revised manuscript.

As suggested by the reviewer, we have added to the Discussion (p13) the importance to carefully assess the safety features of LRY modified TCRs in clinical trials.

Minor points:

1. As indicated already above, many experiments and methods should be explained in greater detail. For each figure, the number of replicates, independent experiments and the statistics should be explained more clearly in a structured manner. Also the hybridoma clones for the main antibodies used for the studies should be mentioned?

REPLY:

We apologize that the information was incomplete. We have added the requested information in the revised manuscript.

2. How was sequencing performed? What is the composition of the sequencing library (how many TCRs from how many donors and so on)? How were alpha-beta pairs of weak and strong TCRs identified (via single cell PCR?).

REPLY:

We have added more information about the TCR library that we used to identify our candidate residues, and we also generated a more extensive library to identify additional candidate residues that might affect TCR expression levels (see also reply to point 1 above). The question about alpha-beta pairs is answered in the reply to reviewer 2, point 3 below.

3. Some of the figures miss axis labelling (e.g. Fig. 1D) or would benefit from legends (e.g. Fig. 4C).

REPLY

We apologize for this; it has been corrected in the revised manuscript.

4. Fig. 4C: Can membrane-bound and cytosolic TCR signal be distinctly quantified?5. In Suppl. 5. In Suppl. Fig. 2 it is mentioned “This is a representative of one individual experiment”; how can this be “representative”?

REPLY

The membrane bound TCR was quantified using mean fluorescence intensity of surface staining data generated by flowcytometry. The cytosolic TCR was studied by confocal microscopy. One of the TCRs was not detectable on the cell surface, which enables the quantification of cytosolic TCR. We agree, that it is more challenging to precisely quantify cytosolic location when the TCR is also expressed on the surface. Nevertheless, our data show that the quantity of cytosolic TCR is independent of the quantity of TCR on the surface.

The images were analysed using Metamorph 7.8 from Molecular Devices. Cells were detected using DAPI nuclear staining and fluorescence was analysed using the Multi Wavelength Cell Scoring module. For each cell this produced an integrated fluorescence value per wavelength (area multiplied by intensity), which was then used to assess expression of CD19 and the TCRs.

We apologize for the error in the legend of supplementary figure 2. It should read that “this is a representative of five experiments”. We have corrected this mistake in the revised manuscript.

Reviewer #2 (Remarks to the Author):

Review for Thomas et al, Framework engineering to produce dominant T cell receptors with enhanced antigen-specific function

A poorly characterized mechanism by which T-cells are able to fine tune their sensitivity is through enhanced TCR avidity by modulating the density of TCR at the cell surface. TCR gene therapy applications looking to exploit this mechanism have focused on increasing the total surface expression and minimizing mispairing between exogenous and endogenous TCR chains through engineering of the $\alpha\beta$ interface, either the TCR transmembrane

domain, C α C β interface or V α V β interface.

Thomas et al have identified a small number of TCR framework mutations which mutate the V α C α or V β C β interface. Surprisingly despite not sitting at the $\alpha\beta$ interface these mutations reduce mispairing between endogenous and exogenous TCR and increased TCR density at the T-cell surface. This translates to a substantial enhancement of antigen specific responses. TCR avidity is often overlooked in the literature which focuses primarily on TCR affinity enhancing mutations. There are two main limitations of the study. Firstly the statistical analysis used to identify TCR framework mutations is limiting and secondly additional experiments are required to support the conclusions.

The authors identify two T-cell populations which may correlate with relative high or low TCR surface density, providing a novel approach to identify candidate framework mutations which may lead to improved surface display. However this dataset is not fully exploited due to the limited statistical analysis used. From tableS1 ~70% of the α framework positions appear to be statistically significant (no data is shown for the β -chain). Differences between the two TCR populations are dominated by a small number of v-genes highly abundant in PBMCs. It is likely that as a result the paired statistical analysis fails to highlight the subtlety of the dataset. As a result the authors have primarily restricted their mutagenesis panel by structural analysis. Whilst this is of merit it undervalues the potential strength of their previously outlined approach.

REPLY

We agree that there are many residues that were significantly different between the dominant and weak TCR library. The structural information was essential to focus the detailed functional analyses on a manageable number of candidate residues.

To explore the point raised by the reviewer about the role of other alpha chain residues, we have performed functional tests of 10 additional residues that were significantly enriched in the dominant TCR library shown in supplementary table 1. We focused on 10 residues T3, S7, P9, A16, T18, S22, D26, Q49, M50 and K90 because they were present in the dominant alpha chains V α 38-1, V α 38-2 and V α 9-2 (Fig. 1d), and rare in the remaining 44 alpha chains of the IMGT database (no more than 2/44 had these residues). We were interested to know whether any of the 10 residues could further enhance the expression of an LRY-modified TCR. The attached figure 3 for the reviewers demonstrates that 7 of the 10 tested residues showed no benefit and that T18, S22, D26 mediated a small improvement in TCR expression. Similar results were obtained with two different TCRs (CMV1-TCR and the Weak1-TCR). This supports our finding that LRY plays a major role in enhancing TCR expression, with further small improvements by the modification of additional residues.

1. There is no analysis of how frequently identified mutants are found in other V-genes. Is the identified mutant due to a reversion to the V-gene consensus at this position or a novel stabilizing mutation limited to few v-genes?

REPLY

We have done this analysis for the important LRY residues. The L96 α is found in 11 out of 47 TCR alpha variable genes of the IMGT database. The RY combination at position 9 and 10 of the beta chain is found in 13 out of 60 variable genes.

We do not think that the residues are mutants, but they do naturally occur in a subset of human TCR variable genes. When we introduce these residues into variable genes that do not naturally have them, this leads to improved TCR expression.

For example whilst it is true that R9 β 10Y β are statistically more common in the high surface expressing TCRs, why are the 13 TRBVs with R9 β 10Y β not enriched in the dominant population? Some R9 β 10Y β V-genes (TRBV 7-7,11-3,25-1,27) appear to be enriched in the weak population!

REPLY

We have carried out experiments with strong and weak α $\square\square\square$ β chain combinations and found that the TCR α chain has a stronger effect on TCR expression levels than the β chain. We have also analyzed a poorly expressed TCR that does have the RY residues in the β chain but lacks a hydrophobic residue at position 96 of the α chain. In this case, the single amino acid change from proline to L96 α is sufficient to increase TCR expression approximately 3-fold (see attached figure 4 for the reviewers). This can explain why some R9Y10 β TCR beta chains may be present in the weak TCR library.

A pairwise analysis of each V-gene dominant or weak, to the consensus V-gene may assist the current statistical model. For example poor surface expression associated with TRAV29 has been attributed to K44 which deviates from the consensus Q44 modulating the $\alpha\beta$ interface. See US patent 20180162922

REPLY

We agree that additional residues may cause subtle effects on TCR expression (see above). We have tested some additional residues as outlined in the text above and in figure 3 for the reviewers.

We have performed additional TCR repertoire analysis as outlined below.

2. Due to the limited sequencing depth most V-genes are poorly represented

REPLY

We have used next-generation sequencing to generate new libraries of dominant and weak TCRs. We used two donors and the cell sorting approach shown in figure 1c to generate a dominant TCR library (23,511 distinct alpha and 38,920 distinct beta clonotypes), and a weak TCR library (29,419 distinct alpha and 41,980 distinct beta clonotypes). We determined the frequency of amino acid residues at each position of the IMGT variable region framework 1, 2 and 3 of the alpha and beta chain. At beta chain position 9 and 10 the amino acids R and Y were significantly enriched in the dominant TCR library compared to the weak TCR library (supplementary Table 1b). At position 96 of the alpha chain, the hydrophobic residue V was enriched in the library generated by next generation sequencing, while L was enriched in the original library. We

performed functional analyses of the hydrophobic residues L, V, I, M and found that they were all effective in enhancing TCR expression levels (see supplementary Fig 3). These additional data confirm an important role of position 96 α where enhanced TCR expression can be mediated by leucine and by biochemically equivalent hydrophobic residues. This is discussed in the result section on page 6 of the revised manuscript.

Statistical analyses of the TCR library generated by next generation sequencing showed that dominant TCRs had over-representation of certain amino acids at 63 of the 77 framework positions of the α chain, and at 68 of 77 positions of the β chain (suppl. Table 1). Of the 63 α chain positions, 51 also showed over-representation in the original TCR library, including the same amino acid in 33 positions. Thus, statistical analyses showed considerable overlap of over-represented residues in the original library and the new library. The analyses also showed that the 3D modeling was essential to focus the functional experiments on a manageable number of candidate residues.

The next generation sequencing method and statistical analyses are described in the Methods section (page 21) of the revised manuscript. The data are described on page 4 and discussed on page 12 of the revised manuscript.

3. TCR sequence data is unpaired, α chain sequences are not linked to their associated β -chains which significantly reduces the analysis that can be done.

REPLY

We agree with the reviewer that unpaired TCR sequence analysis carries certain risks. It assumes that certain V region residues can mediate efficient TCR expression irrespective of the α/β combination. This seems to be the case, as we observed that α variable chain sequences have a stronger effect on TCR expression levels than β variable sequences (see above and Supplemental Figure 11). Although paired analysis may provide additional insights into how α/β pairing affects expression, we were more interested in the identification of residues that can improve TCR expression irrespective of α/β pairing.

4. There is no analysis of J-region usage (linked to $\alpha\beta$ interface stability)

<https://doi.org/10.1016/j.molimm.2008.09.021>

5. There is no analysis of CDR3 length and composition (linked to $\alpha\beta$ interface stability) <https://doi.org/10.1016/j.molimm.2010.01.012>

REPLY

We have performed J-region and CDR3 length analysis on the new TCR library generated by next generation sequencing. The CDR3 lengths (number of amino acids) were similar in dominant TCRs (mean lengths: α 13.69, SD 1.79; β 14.36, SD 1.76) and weak TCRs (mean lengths: α 13.58, SD 1.82; β 14.47, SD 1.90). The J-region analysis showed that 11 out of 61 J-segments of the α chain were significantly over-represented in dominant TCRs and 6 out of 14 J-segments of the β chain were over-represented in dominant TCRs ($p < 0.00001$). This is in agreement with the published observation that the sequence of the CDR3 loop can affect the efficacy of TCR expression (*Blood* **109**, 235-243; 2007). However, the main goal of this study was to identify framework residues that may improve TCR

expression while leaving the CDR loops unchanged.

Additional experiments that would significantly improve confidence in the various conclusions drawn are:

1. Introduced mutations lead to improved TCR stability

Validation required: The thermal stability of the TCR should be assessed (typically measured using DSC or DSF).

REPLY

Our studies have been carried out with full length human TCRs expressed on the cell surface. The suggested DSC or DSF assays require soluble proteins. The generation of soluble TCRs usually involves the introduction of an artificial disulphide bond in the native C α -C β interface and truncation of the membrane proximal part of each constant domain. These structural modifications make it risky to assess the effect of the LRY modifications, which sit at the variable/constant interface and improve TCRs with complete C-domains and no additional disulphide bonds.

The statement in the manuscript that “L96 α is most likely due to enhanced stability of the V α -C α interface” is based on structural modeling. We agree that this statement should be tuned down to reflect the limitations of modeling. We have done this on page 7 of the revised manuscript.

2. Increased TCR expression leads to substantial enhanced T cell potency

Validation required: It should not be assumed that antigen distal residues are unable to impact on the affinity of the TCR for target antigen. At least in antibodies, this would not be unprecedented (<https://doi.org/10.1073/pnas.1613231114>). The affinity of the TCR for antigen (typically measured by BIAcore) should be determined.

REPLY

As outlined above, soluble TCRs have modifications that may interfere with the analysis of the LRY modification. Consistent with a generic effect, the LRY modifications are distal to any CDR loops involved in peptide/MHC recognition, and very unlikely to directly affect interactions at the TCR/peptide-MHC interface. This is consistent with the results obtained with the alanine scanned peptide variant of the CMV1 TCR showing that LRY did not change the fine specificity profile (Fig. 7g). We have extended the alanine scan analysis to two more TCRs, showing that LRY did not change the TCR fine specificity profiles (new supplementary figure 6).

3. “We anticipate the V-region engineering approach we outline can enhance surface expression of most human TCRs”

Validation preferred: Although 8 TCRs have been tested greater confidence would be placed in this statement if different TCRs were selected. As an example at IMGT 96, 10 amino acids are utilised by the 47 TRAVs. However in 6/8 TCRs tested the mutant was P96 α L. Whether Leu leads to improved stability when substituting other amino acids, or indeed if Leu is the best a.a at this position is unknown.

REPLY

We found that the hydrophobic amino acids Isoleucine, Valine and Methionine were also able to improve TCR expression (new supplementary figure 3). Similarly, biochemically equivalent residues at position 9 and 10 of the beta chain have similar effects as arginine and tyrosine at these positions. These data are now included in supplementary figure 3 and discussed on page 6 of the revised paper.

4. Introduced mutations lead to improved surface expression

Validation preferred: Whilst relative intensity MFI clearly demonstrates an improvement it would improve the study if fluorescence intensity could be quantified using an antibody specific standard curve for example using a Quantum™ Simply Cellular kit (Bangs Laboratories, Inc.)

REPLY:

To quantify TCR expression we have used flowcytometry and staining with antibodies against V5, myc and variable regions, and also tetramer staining. All staining reagents have shown that the LRY modifications resulted in improved TCR expression as determined by MFI. We have included additional tetramer staining data indicating that the LRY modification improves expression of correctly paired TCRs. This is shown in figure 6e and f and discussed on page 9 of the revised manuscript.

5. TCR density improves T cell avidity

Validation preferred: Use pHLA multimers to show enhanced staining.

REPLY:

As suggested by the reviewer, in the revised figure 6e,f of the manuscript we have included additional data to show that the LRY modifications increase the percentage of tetramer binding cells by approximately 3-fold, and also the MFI of tetramer-binding (see also point above).

Below are a few minor changes that should be addressed prior to publication:

1. Introduction, line 2: "The ability of engineered T cells to be stimulated by low concentration of peptide antigen is a key parameter for the efficacy of TCR gene therapy. Hence, mutagenesis of the antigen binding regions CDR1, 2, and 3 has been employed to select TCRs with enhanced affinity for the cognate peptide antigen." This doesn't really make sense. Natural T-cells have been shown to activate against very low epitope levels (PMID: 12397360) so why do you need to enhance the affinity? I think the authors are talking about cancer specific TCRs which are often weak affinity and do not enable cancer specific T-cells to control tumours very effectively. This should be outlined and referenced (PMID: 22949370).

REPLY:

This is indeed with reference to cancer-specific TCRs. As suggested by the reviewer, we have made this clear on page 3 of the revised manuscript and we also added the suggested reference.

2. Introduction, line 6: “However, affinity matured TCRs display enhanced cross-reactivity, which can increase the risk of off-target toxicity in patients”. This statement is not accurate and is very misleading. It has not been shown that affinity matured TCRs are more cross-reactive (a sweeping statement with little evidence!). In one of the examples they cite, the affinity matured induces off-target toxicity through a single self-peptide expressed in cardiac tissue. In the second example, the neurological toxicity was attributed to ON-TARGET toxicity because the target antigen was also expressed in the human brain. Additionally, there are several examples of affinity matured TCRs that do not elicit off-target effects (in the clinic and published). The authors should delete this sentence.

REPLY:

We have added new references showing that affinity maturation of TCRs into the supra-physiological range is associated with increased cross-reactivity and loss of specificity for the cognate peptide. We agree with the reviewer that this has not been formally demonstrated in the previously cited references. Accordingly, we have modified the text and references in the revised manuscript.

3. Introduction line 9: “Consequently, TCRs with un-physiologically high affinity fail to trigger T cell responses at low peptide concentrations that are sufficient for triggering robust responses by wild-type TCRs”. The last part of this sentence does not make sense. I assume the authors mean compared to wild-type TCRs with μ M affinities? Alternatively: “Consequently, TCRs with super-physiologically high affinity can fail to trigger T cell responses at low peptide concentrations.”

REPLY:

Yes, the sentence suggested by the reviewer “Consequently, TCRs with super-physiologically high affinity can fail to trigger T cell responses at low peptide concentrations.” captures what we want to say. We have changed this in the revised manuscript.

4. Introduction generally: The introduction is very brief, and quite poorly written, particularly the first paragraph. The authors should consider re-writing the introduction with more consideration for the published literature.

REPLY:

We think that the introduction is concise and covers the prior work that is relevant to this manuscript. Nevertheless, we have followed the reviewer’s suggestions above, and changed the content and references of the first paragraph.

5. The nomenclature for naming of mutants is unconventional. This is particularly problematic for the TCRs in Fig3. Dom TCR: L39a, R55a, Q43b appears to correspond to the WT TCR.

REPLY

We apologize for the confusion in Fig. 3c and d. The residues L39a, R55a,

Q43b are naturally present in the dominant TCR, and the residues F39 α , D55 α and R43 β are naturally present in the weak TCR. We have changed the labels in figure 3 to avoid confusion.

6. The TCR described as wdom TCR F39a, D55a & R43b is actually reverting 3 of the 15 mutations that have been previously introduced back to the germline encoded residues.

REPLY:

This is correct and it has been described in the results section.

7. Standard nomenclature should be used throughout. Mutants should be P96 α L (not P96 α to L96 α mutant or Position 96 alpha proline  leucine). The authors flip between TRAV12-2 and the less conventional format TRAV12.2. In some instances a and b are used instead of α and β .

REPLY:

Apologies for the inconsistency. We have addressed this in the revised manuscript.

8. Conclusions: The authors state: "For some TCRs analysed, a 3-fold increase in TCR density resulted in more than 1000-fold reduction in the peptide concentration that was required to trigger robust T cell effector function." What do they mean by 'a robust T cell effector function'?? They do not define what this is, or show it experimentally. All they can say is that the Emax was higher for INF γ and IL2 release at very high peptide concentrations. The effect is reduced at more physiological peptide concentrations (1-3 nM). EC50s would be more informative.

REPLY:

Please see reply to point 3 of reviewer 1. In figure 7f the response of the LRY-modified CMV1 TCR at 1 nano-mole peptide concentration is greater than the response of the wild type TCR at 10 micro-mole peptide concentration. This is the basis for the statement that more than 1000-fold more peptide is required to trigger an equivalent effector response.

9. Conclusions: The authors state: "The observation that TCR density improves T cell avidity is consistent with previous studies showing that high TCR expression levels were essential for efficient T cell activation by weak agonist peptide." They do not observe that TCR density improves T-cell avidity. They need to show this (tetramers??) to make this claim.

REPLY

We show in the figure 6e,f of the revised paper that the LRY modification improves the percentage and MFI of tetramer binding.

10. Conclusions: "How peptide binding initiates TCR signalling is not fully understood, but proposed mechanisms include serial triggering, kinetic proofreading and conformational change". Additionally, the authors should consider how TCR density might alter triggering via the kinetic segregation

model.

REPLY:

We have added the kinetic segregation model and also added a relevant reference.

11. Figures:

Fig 1A: Label each domain and Cysteine position.

Done

Fig 1B: Label non-transduced in bottom right quadrant (rather than text body)

Done

Fig 1D: Order V-genes according to dominance rather than numerical.

Done

Fig 1E: reformat to be consistent with the clearer SupTable1

Done

Fig2/FigS2: Share the same Fig legend “expressing an endogenous TCR” should be in the title

Done

Fig2C: Label CD19 high/ CD19 intermediate

Done

Fig5: Fig legends within figures are not necessary. Identifying crystal structures of other V-genes which already include the combination of mutations that are being modelled would have assisted in the modeling approach. The labels in the figure are likely going to be too small to read in print. The authors need to remake this figure with consideration to how it will appear in print. It is poor resolution (could just be the PDF) and a bit thrown together at present.

We have changed this figure as suggested

Fig7F: Why does the CVM1 wild-type TCRs fail to generate a dose response to the different peptide concentrations? The response is very similar at all concentrations tested. This would indicate that this TCR just has a lower E_{max} than the LRY modified TCRs and might be sensitive to lower concs of peptide if titrated down further. The authors need to calculate EC_{50} s for these experiments. The HA.m2 wild-type seems to produce no INF γ response to cognate peptide at any dose compared to the irrel peptide. How do the authors explain this? In figure 7A-E this TCR does generate responses to the peptide?

See reply to point 3 of reviewer 1

REVIEWERS' COMMENTS:

Reviewer #1 (Remarks to the Author):

The authors have provided a significant number of additional experimental data in the revised version of the manuscript, which erased most of my concerns.

Reviewer #2 (Remarks to the Author):

Thomas et al have identified several V-genes that are associated with dominant surface TCR expression. From these data three amino acids have been identified that lead to improved TCR presentation and reduced endogenous mispairing. This translated to enhanced avidity mediated antigen-specific T cell responses.

The authors have completed additional work, which has significantly increased the confidence that can be placed in the data. Identifying only a small number of V-genes in the dominant population, such as those shown in Fig 2d it is challenging to understand which a.a in the V-gene are driving the improvement and which positions are just carrier residues. Expanding the depth of the initial analysis, describing the positional statistical analysis used, and accordingly expanding the panel of mutations that were tested provides greater confidence that the hydrophobic residues at position 96 and RY residues highlighted in this study are the primary amino acids responsible for driving enhanced presentation of TCRs.

The overall quality of the text and figures is now of a high standard and the suggested revisions have been satisfactorily addressed.

A couple of minor points that should still be addressed.

The exact layout of Figure 5 is still a little unclear. This data would presumably appear as a single panel in the final publication, it seems likely that some of the text is still much too small/ difficult to read in print.

The new data in Supplementary table 1b is a simplification of a lot of data that would likely be of interest to the reader. As for Figure 1d the V-gene distribution in both the Dom and Weak TCR populations could be shown. The authors could also show increased frequencies at positions in weak TCRs to highlight TCRs in which the LRY mutations are likely to be particularly useful. Finally the frequencies of each amino acid could be reported (potentially as a heatmap for example), to visually demonstrate this data.

Point by Point Reply to reviewers comments

Reviewer #1 (no comments):

Reviewer #2:

The exact layout of Figure 5 is still a little unclear. This data would presumably appear as a single panel in the final publication, it seems likely that some of the text is still much too small/ difficult to read in print.

REPLY

We have supplied figure 5 on one A4 page with readable text

The new data in Supplementary table 1b is a simplification of a lot of data that would likely be of interest to the reader. As for Figure 1d the V-gene distribution in both the Dom and Weak TCR populations could be shown. The authors could also show increased frequencies at positions in weak TCRs to highlight TCRs in which the LRY mutations are likely to be particularly useful. Finally the frequencies of each amino acid could be reported (potentially as a heatmap for example), to visually demonstrate this data.

REPLY

In addition to supplementary table 1, we have provided new supplementary figures 7 and 8, which show a detailed analysis of all amino acids that are significantly overrepresented in dominant TCRs or in weak TCRs. This provides information about which amino acids are associated with strong TCR expression, and which amino acids are associated with weak TCR expression. In order to extend the data in figure 1d, we have added a new supplementary table 2, which shows the relative frequencies of all TCR variable alpha and beta genes in the dominant and weak TCR libraries.